# T1: Tool-integrated Verification for Test-time Compute Scaling in Small Language Models

**Minki Kang**[1]* **Jongwon Jeong**[2]* **Jaewoong Cho**[3]
[1]KAIST  [2]University of Wisconsin-Madison  [3]KRAFTON

## Abstract

Recent studies have demonstrated that test-time compute scaling effectively improves the performance of small language models (sLMs). However, prior research has mainly examined test-time compute scaling with an additional larger model as a verifier, leaving verification by sLMs underexplored. In this work, we investigate whether sLMs can reliably verify the output candidates under test-time scaling. We find that even with knowledge distillation from larger verifiers, sLMs struggle with verification tasks requiring memorization, such as numerical calculations and fact-checking. To address this limitation, we propose **Tool-integrated verification (T1)**, a two-stage framework that first filters candidates with external tools and then uses an sLM for final verification, offloading memorization-heavy steps to tools such as a code interpreter. Within T1, we prove that offloading to external tools reduces the memorization burden on sLMs and improves test-time scaling performance. Experiments on the MATH benchmark demonstrate that, with T1, a Llama-3.2 1B model under test-time scaling outperforms the significantly larger Llama-3.1 8B model. Moreover, T1 improves the verification accuracy of both process reward models (PRMs) and critic models. Our findings highlight the potential of tool integration to substantially improve the verification abilities of sLMs.

## 1 Introduction

Recent advances in large language models (LLMs) have demonstrated strong emergent abilities through large-scale pretraining (Brown et al., 2020; Hurst et al., 2024; Reid et al., 2024), enabling them to tackle complex reasoning tasks such as mathematical problem-solving and competitive coding (Wei et al., 2022b; Lightman et al., 2024). While small language models (sLMs) offer advantages in deployment efficiency and cost (Liu et al., 2024; Lu et al., 2024), they struggle significantly with high-complexity tasks (Wei et al., 2022a).

*Test-time compute scaling* has emerged as a promising approach to enhance sLMs by dynamically allocating additional computation during inference (Wu et al., 2024). Prior works suggest that test-time scaling can surpass pretraining-based scaling (Snell et al., 2024), allowing a 3B LM to outperform a 405B LLM on mathematical benchmarks such as MATH and AIME (Liu et al., 2025). This success depends on reliable verification of generated solutions.

To enable reliable verification, existing approaches have leveraged process reward models (PRMs) and critic models (Wang et al., 2024a; Zhang et al., 2024), but these typically require LLMs (7B+ parameters). Relying on large verifiers counteracts the efficiency benefits of sLMs. Therefore, it remains unclear whether *verification*, where sLMs verify the generated solutions, can enable strong reasoning capabilities without relying on larger models. This raises a key research question:

*Can small language models reliably perform verification for test-time scaling?*

While verification is often easier than generation, prior work has shown that sLMs still struggle to verify the solutions (Song et al., 2024). Our concept-proof experiment in Figure 1 (b) confirms this finding that larger models can reliably verify with chain-of-thought reasoning alone. In contrast,

---

*Equal contribution. Work done at KRAFTON. Contact: minkikang@kaist.ac.kr

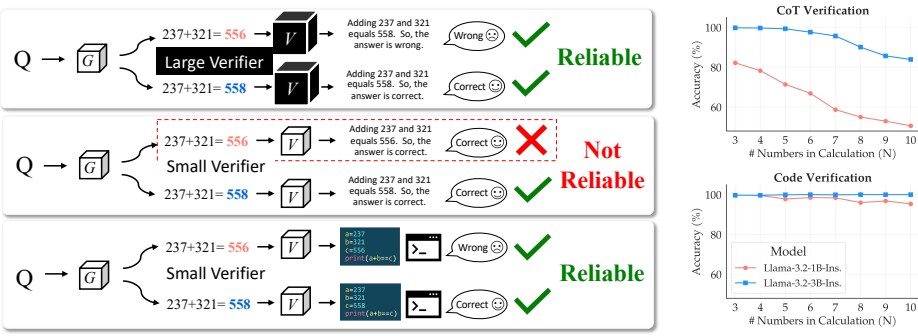

(a) Concept Figure        (b) Concept-Proof Results

Figure 1: **(a) Concept.** Small language models (sLMs) often fail due to their limited capacity. However, when sLMs utilize external tools, their reliability improves. **(b) Concept-Proof Experimental Results.** We evaluate Llama 1B and 3B models on verifying calculations of $N$ three-digit numbers. The 1B model's performance drops significantly as $N$ increases, while the 3B model remains stable. Enabling code generation and execution largely mitigates this drop for the 1B model.

sLMs fail to verify even simple calculations, particularly as the complexity of calculations N increases. We hypothesize that this gap is due to the limited capacity of sLMs to memorize all calculation facts required for verification (Kandpal et al., 2023).

However, as shown in Figure 1 (b), code generation and execution substantially improves sLM's verification accuracy, narrowing the gap with larger models even as $N$ increases. This result suggests that integrating external tools with sLMs reduce the need to memorize arithmetic facts. Therefore, tool integration is not merely beneficial but *necessary* to enable successful verification in sLMs.

Motivated by these findings, we introduce **Tool-integrated Verification (T1)**, a two-stage framework in which external tools first verify candidate solutions and sLMs then verify the filtered solutions. By offloading memorization-heavy steps, such as numerical calculations and fack-checking, to external tools, T1 enables sLMs to achieve verification accuracy comparable to much larger models without increasing parameters. Crucially, this two-stage design applies seamlessly to both generative verifiers and process reward models (PRMs), offering a single framework for test-time scaling. Moreover, the two-stage design theoretically guarantees that tool-based verification reduces the memorization burden and improves test-time scaling performance.

Our experiments demonstrate that T1 enables both generative verifiers (Zhang et al., 2024) and PRMs for more accurate verification. This leads to significant performance improvements on widely-used mathematical and multi-domain reasoning benchmarks, with notable gains on GSM8K (Cobbe et al., 2021) and MATH (Hendrycks et al., 2021). These results underscore the effectiveness of T1 in improving the performance of sLMs under test-time compute scaling.

Our contributions are as follows:

- We conduct a systematic study of sLMs' verification under test-time scaling, identify memorization-heavy steps in verification as a key bottleneck, and motivate addressing them with external tools.
- We propose Tool-integrated Verification (T1), a two-stage framework that leverages external tools to offload memorization-heavy steps before sLM verification.
- We provide a theoretical analysis showing that external tools reduces the memorization burden and that two-stage design in T1 improves test-time scaling performance.
- We show that T1 integrates seamlessly with both generative verifiers and PRMs, achieving strong results on math reasoning benchmarks GSM8K and MATH.

## 2 RELATED WORKS

### 2.1 TEST-TIME COMPUTE SCALING

Test-time compute scaling has emerged as a promising approach for improving the reasoning capabilities of large language models (LLMs) (Wu et al., 2024). It can be broadly categorized into

*sequential* and *parallel* methods (Snell et al., 2024). Sequential scaling iteratively refines solutions by leveraging post-training, enabling the model to perform self-reflection and verification (Muennighoff et al., 2025; DeepSeek-AI et al., 2025). Parallel scaling, in contrast, generates multiple candidate solutions simultaneously and selects the best one using a verifier model (Cobbe et al., 2021; Lightman et al., 2024; Brown et al., 2024). A common strategy is the *best-of-N* method, which produces $N$ parallel outputs and ranks them based on verification scores (Cobbe et al., 2021; Lightman et al., 2024). Increasing $N$ has been shown to enhance LLM on challenging benchmarks (Brown et al., 2024; Snell et al., 2024).

In this work, we focus on the *parallel scaling* paradigm due to its simplicity and popularity. Prior research shows that even small models can achieve strong results when paired with a large verifier in parallel scaling (Liu et al., 2025). We further investigate whether small language models (sLMs) can verify, enabling test-time scaling without large models.

## 2.2 VERIFIER IN TEST-TIME COMPUTE SCALING

The verifier plays a crucial role in parallel scaling. One approach, the Process Reward Model (PRM), scoring each reasoning step individually using a regression head, enables fine-grained feedback (Lightman et al., 2024; Wang et al., 2024a; Zeng et al., 2025). An alternative approach leverages an LLM itself as a *Critic* model, prompting it to evaluate reasoning steps (Zheng et al., 2023). Zheng et al. (2024) have shown that powerful LLM-based critic models can outperform PRM, particularly in mathematical reasoning tasks. Recent works (Zhang et al., 2024; Mahan et al., 2024) proposed the Generative Reward Model (GenRM), formulating verification as a next-token prediction problem with chain-of-thought (Wei et al., 2022b) improving interpretability in step-wise verification.

Despite these advances, ensuring consistent and high-quality step-wise verification remains an open problem for both PRM and GenRM. Additionally, prior works have not thoroughly explored the change in verification performance depending on the size of LMs.

## 2.3 TOOL-INTEGRATED LANGUAGE MODEL

The integration of external tools has significantly enhanced LLM capabilities. Program-aided language models (Gao et al., 2023) introduced delegating computations to interpreters via synthesized code. Subsequent works expanded this by using tools like search engines and calculators for fact retrieval and arithmetic (Schick et al., 2023; Qin et al., 2024). Recent methods further integrate tools into multi-step reasoning (Gou et al., 2024b; Zhu et al., 2024), reward modeling (Li et al., 2024), and self-correction (Gou et al., 2024a).

Our work extends this line of research by studying **how to design tool integration that benefits sLM verification across both process reward models (PRMs) and critic models**. We formulate tool use as an additional dimension of test-time scaling, emphasizing its effectiveness in memorization-heavy verification tasks.

## 3 PRELIMINARIES

**Test-time scaling** Following Snell et al. (2024), we view test-time scaling as modifying the model's proposal distribution. Given a problem $x \in \mathcal{X}$, we sample a solution $y$ from the policy $\pi(y \mid x, I_p; \theta)$ in the set $\mathcal{Y}$, where $I_p$ is the generator-specific instruction prompt, and the policy is parameterized by $\theta$ which refers to the pre-trained language models.

Several algorithms can be used to scale test-time computation, including those that adjust the input level (e.g., self-reflection (Kumar et al., 2024)) and those that modify the output level (e.g., best-of-N (Cobbe et al., 2021), beam search (Yao et al., 2023)). Among them, the best-of-N algorithm is a simple yet powerful approach. It samples multiple candidate solutions from the policy and selects the one with the highest score, as determined by the verifier, as the final prediction. Formally, the Best-of-N policy $\pi^N(y \mid x, I_p; \theta)$ is defined as:

$$\underset{y \in \{y_1, \ldots, y_N\}}{\arg\max} \ r(x, y), \quad \text{s.t.} \quad y_i \sim \pi(y \mid x, I_p; \theta), \tag{1}$$

where $r(x, y) : \mathcal{X} \times \mathcal{Y} \to \mathbb{R}$ is a verifier that assigns a scalar score $r$.

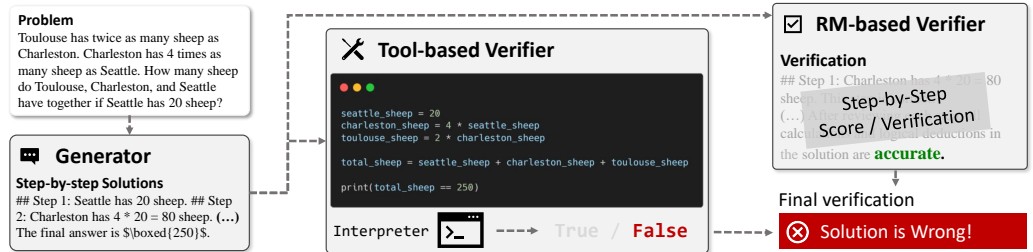

Figure 2: **Tool-integrated self-verification for mathematical reasoning.** (a) **Generator:** A small language model (sLM) may produce incorrect solutions due to calculation errors. (b) **Tool-based Verifier (ToolV):** The sLM generates executable code based on its reasoning; the output of the code is used to verify the solution's correctness. (c) **Reward Model (RM)-based Verifier:** The reward model (GenRM / PRM) still evaluates the solution as before, but its verdict only contributes to the final decision if the solution passes the tool-assisted filter. Concrete examples are in Appendix F.

**Verifier** The verifier can be modeled using the following models: (1) process reward model (PRM), which assigns the score of each step of reasoning (Wang et al., 2024a), (2) critique model, which generates a rationale for the verification (Zheng et al., 2023). For both cases, the sequence of verification scores or tokens $z$ are sampled from $\pi(z \mid x, y, I_v; \theta)$ where $I_v$ is the verifier-specific instruction prompt. In general, we use the last score or token of the sequence as the final score for the solution (Snell et al., 2024; Zhang et al., 2024). For instance, in generative verifier (Zhang et al., 2024), the verification score can be obtained as follows:

$$r(x, y) = \pi(z_T = \text{'Yes'} \mid x, y, I_v, z_{1:T-1}; \theta), \quad \text{s.t.} \quad z_{1:T-1} \sim \pi(z \mid x, y, I_v; \theta), \quad (2)$$

where $z_{1:T-1}$ is chain-of-thought (Wei et al., 2022b) and last token $z_T \in \{\text{'Yes'}, \text{'No'}\}$.

# 4 METHOD

## 4.1 TOOL-INTEGRATED VERIFICATION

Test-time scaling can improve a base policy model that generates valid solutions from its proposal distribution, but the effectiveness of scaling at test-time heavily relies on the performance of the verifier. However, sLMs often struggle to reliably verify the correctness of their generated outputs (Song et al., 2024). Specifically, sLMs exhibit limitations in precisely validating numerical computation or detecting incorrect or outdated knowledge information, due to their limited parameter size and insufficient memorization capacity.

To address these limitations, we propose a **Tool-integrated verification (T1)** approach for parallel test-time scaling in sLMs. As shown in Figure 2, our verification approach involves two stages: 1) filtering stage with **Tool-based Verifier (ToolV)** and 2) scoring stage with reward model (RM)-based verifier, and formally can be expressed as:

$$y^* = \underset{y \in \{y_1, \ldots, y_N\}}{\arg\max} \; f(x, y; \mathcal{T}, \theta) \times r(x, y; \theta), \quad \text{s.t.} \quad y_i \sim \pi(y \mid x, I_p; \theta), \; \forall i \in \{1, \ldots, N\}, \quad (3)$$

where $f(x, y; \mathcal{T}, \theta) \in \{0, 1\}$ indicate a binary tool-based verification function (with 0 indicating a filtered-out response), $\mathcal{T}$ denotes the utilized tool (e.g., code interpreter, retriever), and $r(x, y; \theta)$ denotes the verifier score defined in Equation 1.

**Tool-based verifier stage** In this stage, sLM utilizes the external tool $\mathcal{T}$, such as a code interpreter or knowledge retriever, to verify generated outputs. Here, we assume the scenario in which the utilized tool $\mathcal{T}$ is explicitly known. Specifically, sLM uses these tools to verify numerical accuracy and validate the knowledge in generated solutions. One specific example is that multiple generated responses are filtered based on tool-based verifiers, discarding those with incorrect calculations or inaccurate knowledge information. Specifically, the tool-based verification function, $f(x, y; \mathcal{T}, \theta)$, consists of three parts: 1) generating the tool-calling query (i.e., $c_1$), 2) the execution of the tool (i.e., $\mathcal{T}(\cdot)$), and 3) extraction of the verification (i.e., $c_2$). Therefore, $f(x, y; \mathcal{T}, \theta)$ can be represented as:

$$f(x, y; \mathcal{T}, \theta) = c_2 \sim \pi(c \mid \mathcal{T}(c_1), x, y, I_f; \theta), \quad \text{where} \quad c_1 \sim \pi(c \mid x, y, I_c; \theta), \quad (4)$$

and $\boldsymbol{I}_f$ and $\boldsymbol{I}_c$ are task-specific instruction prompts. Detailed formulation for mathematical reasoning and knowledge-intensive tasks are represented in Appendix B.

**RM-based verifier stage** Following ToolV stage, the remaining generated responses are scored using a reward model, the same model used for generation and filtering. This reward model assesses the overall logical consistency, coherence, and correctness of each response. The final output is chosen as the response with the highest reward score.

## 4.2 VERIFIER DISTILLATION

To further enhance the performance of both verification stages, we employ knowledge distillation (Hinton et al., 2015; Kim & Rush, 2016) from LLMs. Specifically, we fine-tune sLM using tool-based and RM-based verifications generated by a larger teacher model $\theta_T$. To efficiently manage multiple distinct tasks during the distillation process, we adopt a multi-LoRA (Hu et al., 2022) approach, assigning separate LoRA adapters, $\Delta\theta_{\text{tool}}$, and $\Delta\theta_{\text{reward}}$, for each verifier.

The distillation for ToolV is formulated as:

$$\mathcal{L}_{\text{tool}}(\Delta\theta_{\text{tool}}) = -\mathbb{E}_{\boldsymbol{x}\sim\mathcal{X}_{\text{train}},\ \boldsymbol{y}\sim\pi(\cdot|\boldsymbol{x},\boldsymbol{I}_p;\theta),\ \boldsymbol{c}\sim\pi(\cdot|\boldsymbol{t},\boldsymbol{x},\boldsymbol{y},\boldsymbol{I};\theta_T)} \log\pi(\boldsymbol{c}\mid\boldsymbol{t},\boldsymbol{x},\boldsymbol{y},\boldsymbol{I};\theta+\Delta\theta_{\text{tool}}), \quad (5)$$

where $\mathcal{X}_{\text{train}}$ is training dataset, $\boldsymbol{c}\in\{\boldsymbol{c}_1,\boldsymbol{c}_2\}$, $\boldsymbol{I}\in\{\boldsymbol{I}_c,\boldsymbol{I}_f\}$, and $\boldsymbol{t}$ is empty $\phi$ or the output of $\mathcal{T}(\boldsymbol{c}_1)$. Note that we generate tool-based verifications using the teacher model by applying corresponding instruction $\boldsymbol{I}$ in a zero-shot manner (Ouyang et al., 2022; Chung et al., 2024).

Similarly, distillation for the RM-based verifier is expressed as:

$$\mathcal{L}_{\text{reward}}(\Delta\theta_{\text{reward}}) = -\mathbb{E}_{\boldsymbol{x}\sim\mathcal{X}_{\text{train}},\ \boldsymbol{y}\sim\pi(\cdot|\boldsymbol{x},\boldsymbol{I}_p;\theta),\ \boldsymbol{z}\sim\pi(\cdot|\boldsymbol{x},\boldsymbol{y},\boldsymbol{I}_r;\theta_T)} \log\pi(\boldsymbol{z}\mid\boldsymbol{x},\boldsymbol{y},\boldsymbol{I}_r;\theta+\Delta\theta_{\text{reward}}), \quad (6)$$

where $\boldsymbol{I}_r$ is RM-based verifier-specific instruction. In Equation 6, responses are first sampled from the student model's proposal distribution. Each sampled response is then verified by the teacher model, and finally, the student model is fine-tuned based on these verifications.

## 5 THEORETICAL ANALYSIS

In this section, we present theoretical analyses of two aspects with conceptual illustrations: (1) how external tools reduce memorization requirements and improve verification performance (subsection 5.1), and (2) how our two-stage approach guarantees improved performance under test-time scaling (subsection 5.2).

## 5.1 MEMORIZATION BOUND WITH & WITHOUT TOOL

Let us consider a simple verification task in which a verifier assesses whether the given equation $a + b = c$ is true or not. Let this task's data distribution is $q$ and $P = ((a,b,c),r) \sim q$. Assume we have $|\mathcal{X}|$ number of training samples such that $\mathcal{X} = \{((a,b,c),r) \mid a,b \in \{0,\ldots,M-1\},\ c \in \{0,\ldots,2M-2\},\ r = \mathbf{1}_{a+b=c}\}$, where $((a,b,c),r)$ is independently sampled according to $q$. Then, let $X \sim q^{\otimes|\mathcal{X}|}$ be the random variable representing the distribution of the training set $\mathcal{X}$. A learning algorithm $\mathcal{A}$ receives $X$ to produce $\theta$ such that $\theta = \mathcal{A}(X)$. Then, we call that $\mathcal{A}$ is $\varepsilon$-*close-to-optimal* if $\text{err}_{q,|\mathcal{X}|}(\mathcal{A}) \leq \text{err}_{q,|\mathcal{X}|}(\mathcal{A}_{\text{OPT}}) + \varepsilon$, where $\text{err}_{q,|\mathcal{X}|}(\mathcal{A}) = \Pr_{X\sim q^{\otimes|\mathcal{X}|},((a,b,c),r)\sim q,\hat{r}\sim\pi(\boldsymbol{c}|a+b,c,\boldsymbol{I};\theta=\mathcal{A}(X))}(\hat{r} \neq r)$ and $\mathcal{A}_{\text{OPT}}$ is the optimal learning algorithm. Then, Lemma 5.1 shows how much information of $X$ should be memorized within $\theta$ to satisfy almost zero error.

**Lemma 5.1** (Memorization without Tool (Brown et al., 2021))**.** *Any learning algorithm $\mathcal{A}$ that is $\varepsilon$-close-to-optimal with sufficiently small $\varepsilon > 0$ also satisfies $I(X;\theta\mid P) = \Omega(M^3)$, where $I$ is the mutual information.*

*Proof sketch.* Theorem 1.1 in Brown et al. (2021) said that $I(X;\theta\mid P)$ is proportional to at least dataset size. Since $|\mathcal{X}| = 2\cdot(M-1)^3$, we can get $\Omega(M^3)$. Refer to subsection C.2 for the detailed proof. $\square$

On the other hand, using an external tool that verifies whether $a + b = c$ allows the model to avoid memorizing the full table of sums. Specifically, define a tool $\mathcal{T}$. Suppose $\theta$ is generated by learning

algorithm $\mathcal{A}$ that has access to $\mathcal{T}$, and that $f(a, b, c; \theta, \mathcal{T}) = \mathbf{1}_{a+b=c}$ holds. Then we obtain the following result:

**Theorem 5.2** (Memorization with Tool). *Suppose $\theta$ is generated by learning algorithm $\mathcal{A}$ that has access to $\mathcal{T}$, and that $f(a, b, c; \theta, \mathcal{T}) = \mathbf{1}_{a+b=c}$ holds. Then, any learning algorithm $\mathcal{A}$ that is $\varepsilon$-close-to-optimal with sufficiently small $\varepsilon > 0$ also satisfies $I(X; \theta \mid P) = 0$, where $I$ is the mutual information.*

*Proof sketch.* As the learning algorithm can access an external tool $\mathcal{T}$, then $\theta = \mathcal{A}(X)$ such that $f(a, b, c; \theta, \mathcal{T}) = \mathbf{1}_{a+b=c}$. $f$ makes $\mathrm{err}_{q, |\mathcal{X}|}(\mathcal{A}) = 0$. Also, $\theta$ is independently determined regardless of $X$, resulting in $I(X; \theta \mid P) = 0$. See subsection C.3 for the detailed proof. $\square$

$I(X; \theta \mid P)$ quantifies the amount of information about $X$, drawn from $P$, that must be memorized in $\theta$ learned by $\mathcal{A}$ to achieve near-zero error. By comparing $I(X; \theta \mid P)$ from Lemma 5.1 and Theorem 5.2, we demonstrate that a tool drastically reduces the required memorization of $X$ within $\theta$, lowering $I(X; \theta \mid P)$ from $\Omega(M^3)$ to 0. Consequently, this result implies that with the tool, small models become reliable for the verification task.

## 5.2 EFFECT OF TOOL-BASED VERIFIER ON TEST-TIME SCALING

We employ the toy setting introduced in Beirami et al. (2024). Specifically, for given input $\boldsymbol{x}$, the ground-truth label produced by this generator is set as 1, and the generator $\pi$ produces binary outputs, i.e., $\mathcal{Y} = \{0, 1\}$. Furthermore, we consider an imperfect verifier inducing noise. Then, we can show Theorem 5.3 that increasing $q_1$ directly increases the probability of obtaining a correct output from the best-of-$N$.

**Theorem 5.3** (Monotonicity of Imperfect Verifier). *Let the generator output $0$ or $1$ with equal probability, i.e., $\pi(0|\boldsymbol{x}) = \pi(1|\boldsymbol{x}) = \frac{1}{2}$, and the verifier $r$ with noise level $p, q$ be defined as follows:*

$$r(\boldsymbol{x}, 0) = \begin{cases} 0, & w.p. \ p, \\ 1, & w.p. \ 1 - p, \end{cases} \qquad r(\boldsymbol{x}, 1) = \begin{cases} 1, & w.p. \ q, \\ 0, & w.p. \ 1 - q, \end{cases}$$

*with the condition that $p > 1 - p$ and $q > 1 - q$. Assume $\bar{p}$ and $\underline{p}$ be the noise level of two verifiers with $\bar{p} > \underline{p}$. Then, for any $N \geq 2$,*

$$\pi^N(1 \mid \boldsymbol{x})\Big|_{p=\bar{p}} > \pi^N(1 \mid \boldsymbol{x})\Big|_{p=\underline{p}}. \tag{7}$$

*Proof sketch.* By the law of total probability, we get $\pi^N(1 \mid \boldsymbol{x})$. Then, we can get the monotonicity of $\pi^N(1 \mid \boldsymbol{x})$. Refer to subsection C.4 for the detailed proof. $\square$

If tool-based verification function $f$ in subsection 4.1 effectively acts as a filter for incorrect solutions, we can say that using $f$ increases $p$, thus improving the verifier's capability to choose the correct label as shown in Theorem 5.3.

## 6 EXPERIMENTS

### 6.1 SETUP

**Datasets** We mainly focus on **mathematical reasoning** task due to its widespread adoption and strong effectiveness in assessing the reasoning capabilities of language models (Snell et al., 2024; Liu et al., 2025). We use (1) **MATH500** (Hendrycks et al., 2021; Lightman et al., 2024), a dataset containing college-level math problems. (2) **GSM8K** (Cobbe et al., 2021), which consists of grade-school math problems. We use the training set of each dataset for distillation. We also include additional experimental results on the subset of MMLU-Pro (Wang et al., 2024b) in Appendix E, which contain a multi-domain knowledge-intensive problems.

**Evaluation setting** Following previous works (Cobbe et al., 2021; Lightman et al., 2024; Snell et al., 2024; Liu et al., 2025), we evaluate **weighted Best-of-N** performance, where we aggregate the score of the solutions ending with the same final answer, to assess test-time compute scaling. We generate 64 solutions using a fixed generator and measure the percentage of correctly solved problems after verifications. As a verifier, we use both PRM (Wang et al., 2024a) and GenRM-CoT (Zhang et al., 2024) (we refer to it as GenRM).

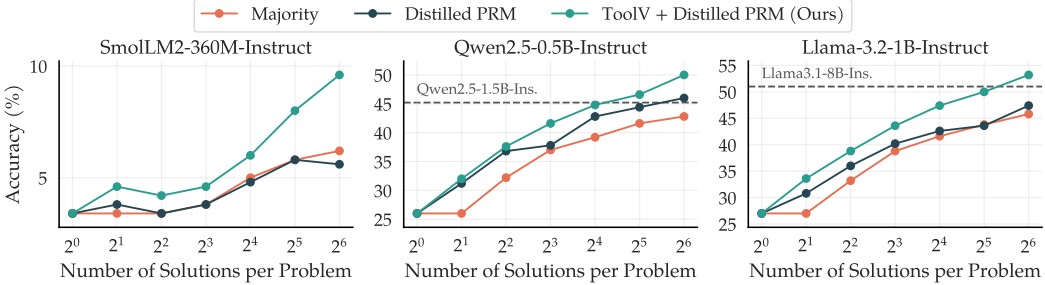

Figure 3: **MATH500 with PRM.** Weighted Best-of-N performance of three small language models, emphasizing the benefits of ToolV on college-level math problems. ToolV significantly enhances PRM, enabling small models to outperform or match much larger models. Qwen2.5-1.5B and Llama3.1-8B performances are reported as $N = 1$ greedy decoding.

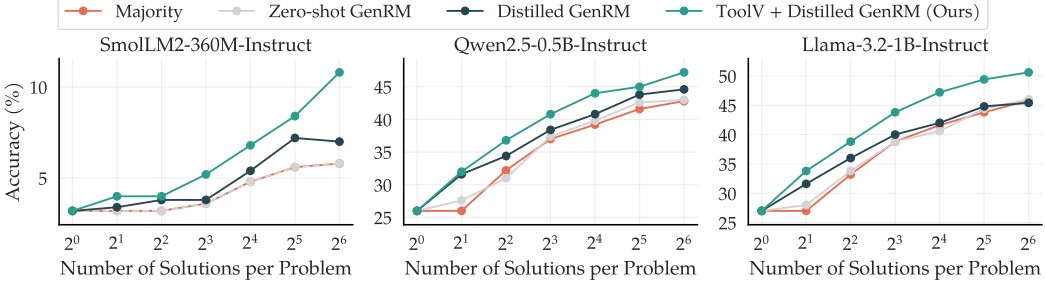

Figure 4: **MATH500 with GenRM.** Weighted Best-of-N performance of three small language models, showcasing the effectiveness of ToolV with GenRM, where even generative verification cannot supplement the calculation error which can be easily filtered out by using a tool.

**Baselines**  We compare ours, **Tool-integrated verification**, that utilizes both fine-tuned reward model and tool-based verifiers (**ToolV**), against the following baselines: (1) **Majority Voting** (Wang et al., 2023) (without using verifier), (2) **Zero-shot GenRM** (Zheng et al., 2023; Song et al., 2024) (without any fine-tuning), (3) **Distilled PRM/GenRM** (with fine-tuning), (4) **Themis** (Li et al., 2024).

**Models & training**  We experiment with the smallest instruction-tuned models from widely used families: Qwen-2.5-0.5B-Instruct (Yang et al., 2024) and Llama-3.2-1B-Instruct (Dubey et al., 2024). In addition, we test SmolLM2-360M-Instruct (Allal et al., 2025) for evaluation in extremely small model. As the teacher model, we employ gpt-4o-mini-2024-07-18 (Hurst et al., 2024). The teacher model is prompted to generate outputs used to fine-tune student models (Hu et al., 2022). For PRM's teacher, we use Qwen2.5-Math-PRM-7B (Zhang et al., 2025).

We include more implementation details in Appendix D.

## 6.2 EXPERIMENTAL RESULTS

**ToolV improves PRM in small LMs**  As shown in Figure 3, ToolV improves performance when combined with the distilled Process Reward Model (PRM) on the MATH500 benchmark. Our results show that adding ToolV provides substantial gains in test-time scaling, suggesting that distilled PRM alone is still prone to numerical errors. Notably, with ToolV, **only using Llama 1B models outperforms the performance of the 8B model**—demonstrating that extra test-time computation can meaningfully boost smaller models, where distilled PRM alone cannot enable the 1B model to reach that performance until generating 64 solutions. Similarly, ToolV enables Qwen2.5 0.5B to match the performance of the 1.5B model by generating just 16 solutions, showing impressive effectiveness.

**ToolV improves GenRM in sLMs**  As shown in Figure 4, ToolV boosts test-time scaling for three small language models on MATH500 when combined with the distilled GenRM (Zhang et al., 2024). While GenRM struggles alone, ToolV compensates—at the cost of code generation. Similar gains appear on GSM8K in Figure 5, especially for SmolLM2-360M-Instruct, the weakest model. This

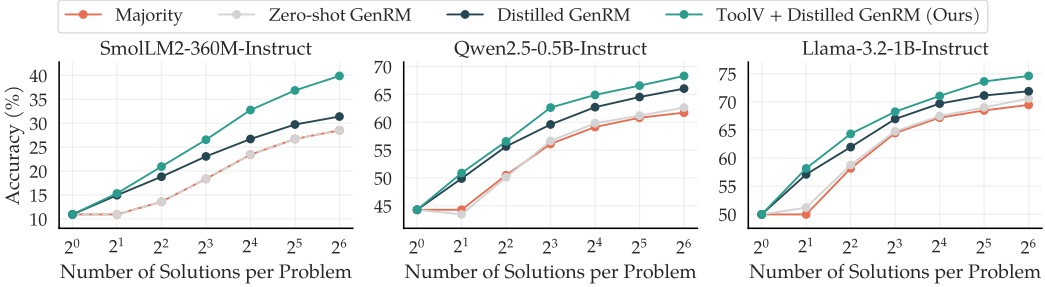

Figure 5: **GSM8K with GenRM.** Weighted Best-of-N performance comparison across three small language models. The results show that ToolV also improves model performance on graduate-level arithmetic problems. However, the gains are smaller on this simpler task, where existing verifiers already perform reliably compared to more challenging tasks.

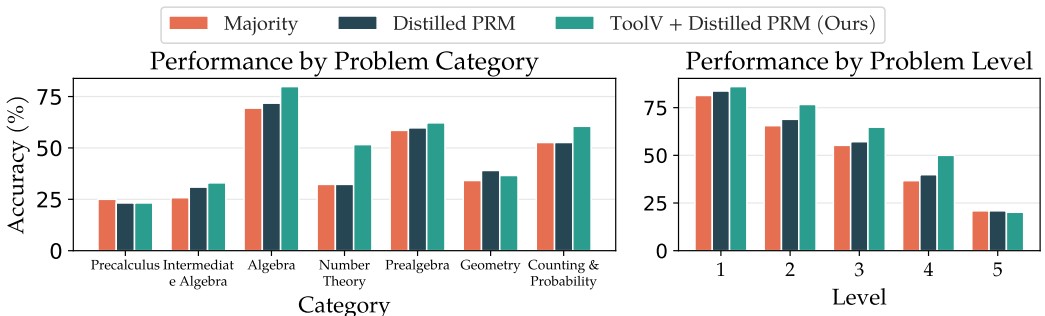

Figure 6: **Analysis with problem types and levels.** We perform analysis on the effect of tool-based verifier with problem types and levels in **MATH500** dataset. The results are from Llama-3.2-1B-Instruct with PRM using weighted Best-of-N ($N = 64$). This analysis shows ToolV is most effective on mid-level problems and calculational domains.

supports our analysis in subsection 5.1 that ToolV enables even small models to memorize key information. Zero-shot GenRM ablations confirm that without distillation, small models struggle to verify solutions (Song et al., 2024).

**Our two-stage method outperforms prior tool-integrated verification.** Themis (Li et al., 2024) demonstrates that integrating external tools can enhance verification performance of 7B models across tasks requiring tools such as calculator, weather, or calendar. In contrast, our focus is on mathematical reasoning benchmarks (e.g., GSM8K, MATH500) and much smaller models that were not addressed in Li et al. (2024). However, it is notable that our *two-stage* tool-integrated verification approach proves effective for small models on math reasoning, whereas Li et al. (2024) explored a unified tool-augmented reward modeling framework.

Similar to our method, we first generate tool-integrated verification trajectories using gpt-4o and distill them into the Llama-3.2-1B-Instruct model. These trajectories are similar to GenRM but explicitly include intermediate Python

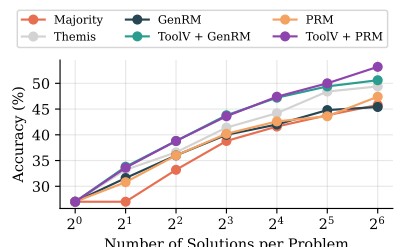

Figure 9: **MATH with Themis as baseline.** Weighted Best-of-N performance of Llama-3.2-1B-Instruct model. ToolV outperforms Themis (Li et al., 2024).

code generation (Gou et al., 2024b). As shown in Figure 9, Themis (Li et al., 2024) surpasses other distilled GenRM and PRM baselines without tool usage. However, both ToolV + GenRM and ToolV + PRM outperform Themis, indicating that our two-stage approach is better suited for test-time scaling on math reasoning with small models, as it can be combined with distilled PRM and ensures performance improvements as analyzed in subsection 5.2.

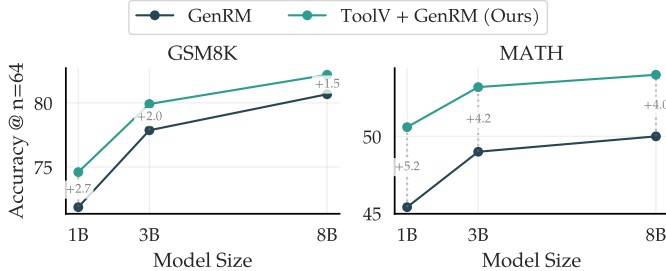

Figure 7: **Effects of ToolV on sizes of GenRM.** Weighted Best-of-N ($N = 64$) performance of GenRM based on different sizes of Llama 3 (Dubey et al., 2024) on **MATH500**. For ToolV, we use 1B and only scale up the GenRM.

Figure 8: **Correct solutions ratio** among $N = 64$ generations to show how the tool-based verifier works.

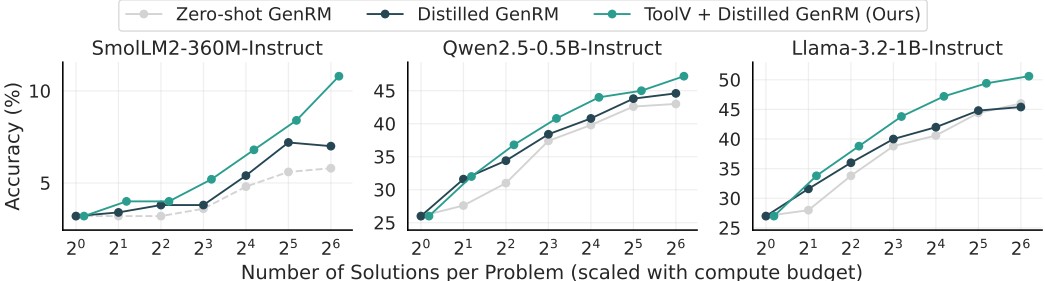

Figure 10: **MATH500 with GenRM under a scaled x-axis reflecting compute budget.** Our method remains the best even when compute budget is taken into account.

### 6.3 ANALYSIS

**Effects of ToolV on category and difficulty** In Figure 6, we analyze the $N = 64$ weighted Best-of-N performance on MATH500 using Llama-3.2-1B-Instruct. On the left, category-wise results show ToolV brings clear gains, especially in Algebra, Number Theory, and Counting & Probability. Geometry sees a drop, likely due to ToolV being less effective in that domain. On the right, performance by problem level shows consistent improvements with ToolV for Levels 2–4. However, results dip at Level 5, suggesting ToolV struggles with the most challenging problems. Overall, ToolV works best on mid-level problems and math areas requiring accurate calculation, but improvements are needed for higher-difficulty cases.

**ToolV benefits larger verifiers** Figure 7 shows how performance varies with distilled GenRM size, keeping ToolV fixed at 1B. As GenRM scales, the gap with and without ToolV narrows but remains. Notably, on MATH500, **1B GenRM + ToolV outperforms 8B GenRM**, suggesting ToolV can be more effective than scaling the verifier—especially on harder tasks.

**Effects of ToolV as filter** Figure 8 shows how ToolV acts as an effective filter for mathematical solutions. Using Llama-3.2-1B-Instruct with GenRM on MATH500 ($N = 64$ candidates per sample), we recalculated accuracy after applying ToolV to remove wrong outputs. The results support our analysis in subsection 5.2, showing ToolV reliably filters out incorrect solutions and significantly improves accuracy.

### 6.4 DISCUSSION: COMPUTATIONAL OVERHEAD OF TOOLV

ToolV requires generating executable code in addition to producing solutions from the generator and verifier outputs such as GenRM. Understanding the compute budget is therefore important to assess the effectiveness of each method and to determine which approach is preferable under constrained compute resources (Singhi et al., 2025). We begin by analyzing the number of generated tokens required by the generator, the

| Stage | Tokens |
|---|---|
| Solution (Generator) | 574.39 |
| Verification (GenRM) | 4431.11 |
| Code (ToolV) | 610.84 |

Table 1: Token usage per stage.

verifier (GenRM), and ToolV on the MATH benchmark using Llama-3.2-1B-Instruct. Table 1 reports the average token count per solution.

As shown, ToolV introduces extra code generation tokens. This raises two natural questions: (1) How can we ensure that the performance improvement comes from ToolV itself, rather than from increased compute? If additional budget is available, why not simply use a larger verifier? (2) Under an equal compute budget, does ToolV still provide benefits compared to using GenRM or PRM alone? We address these questions below.

**Using ToolV is more beneficial than increasing verifier size.**  ToolV enables the effective use of small models, which offers a practical advantage over larger verifiers in terms of GPU memory requirements. Our framework allows small models to function as strong verifiers, which is particularly valuable in memory-constrained environments such as on-device or limited GPU setups.

Even when accounting for the extra compute, the cost of ToolV corresponds to using only a 1.14 times larger verifier. Using Table 1, let $k$ denote the relative scale factor. Then, $k = (5616.34 - 574.39)/4431.11 = 1.14$. Since ToolV 1B + GenRM 1B surpasses GenRM 8B on MATH (Figure 7), the performance gain from ToolV more than justifies this small additional cost, especially when compared with simply scaling up the verifier.

**ToolV still improves performance under the same compute budget.**  To account for the overhead of ToolV, we shift the x axis in performance plots such as Figure 4. Applying the scaling factor of 1.14, we shift the x-axis of our method with GenRM accordingly. The resulting comparison in Figure 10 shows that even after budget normalization, ToolV continues to provide meaningful gains, particularly for the smallest model, SmolLM2-360M-Instruct, and in settings with the large number of generated solutions.

Additional discussion and extended experimental results are provided in Appendix E.

## 7    CONCLUSION

In this work, we introduced **Tool-integrated Verification (T1)**, which delegates memorization-intensive tasks in verification to external tools for sLMs. Our method involves a tool-based verification stage and a reward-model-based scoring stage, both enhanced by knowledge distillation from large verifiers. Theoretical analysis confirmed that tool use substantially reduces the memorization burden on sLMs and improves test-time scaling accuracy. Empirical experiments demonstrated that T1 significantly improves the test-time scaling performance of sLM on mathematical reasoning and knowledge-intensive tasks. A key conclusion of our work is that tool integration is essential for enhancing sLM performance, even under test-time scaling, by reducing the memorization burden.

**Limitations & Future Works**   While T1 shows strong improvements, some limitations remain. (1) ToolV acts only as a rejection filter and cannot recover from false negatives—correct solutions mistakenly rejected by the verifier. As one possible implementation of T1, this limitation could be mitigated by integrating tool-based reasoning into the verification step, allowing the verifier to leverage correctness guarantees from tool outputs (Gou et al., 2024b; Li et al., 2024), which we do not explore in this work. (2) Our work focuses on best-of-N (parallel) test-time scaling, which lacks information sharing between generations. However, tools can also benefit other test-time scaling strategies, such as step-level search (Yao et al., 2023) in sLMs or long reasoning chains in sequential test-time scaling as demonstrated by Li et al. (2025). Exploring these directions presents a promising avenue for future work.

### ETHICS STATEMENT

This work uses only public datasets for math and knowledge tasks and does not involve human subjects or personal data. Tool based verification runs a code interpreter for numeric checks and a retriever over Wikipedia abstracts for factual checks, and these tools do not store user data. The main risks are retrieval errors and code execution failures; we bound both with conservative rules, simple evaluation code, and we discard unverifiable claims. We follow all dataset and model licenses, disclose model families, teacher models, and training compute, and report the resources used.

REPRODUCIBILITY STATEMENT

We specify datasets, baselines, metrics, and the evaluation protocol for weighted Best of N with both a process reward model and a generative verifier. The appendix provides implementation details, training setup, key hyperparameters, and the exact prompts for code based math checks and document based fact checks. We state the retriever, source, and document count per query, and we define the rules for code execution and success signals. We will release an anonymous package with scripts, prompts, configuration. Plus, we will provide instructions and seeds to recreate all tables and figures.

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

# A CONCEPT-PROOF EXPERIMENT DETAILS

We provide additional details about the proof-of-concept experiment shown in Figure 1 (b) of the main paper. The verification task focuses on arithmetic calculations involving randomly selected $N$ three-digit numbers, using both addition and subtraction, with $N$ ranging from 3 to 10.

For each value of $N$, we generate 500 equations with correct answers, along with another 500 equations where the output is slightly incorrect—within a $5\%$ margin of error.

We then prompt Llama-3.2-1B-Instruct (Dubey et al., 2024) to verify these calculations. Specifically, we use Prompt A.1 to make the language model to generate a step-by-step explanation in natural language.

---

**Prompt A.1: Check Calculation**

Evaluate the below calculation. Is this calculation correct? If correct, return True. Return False otherwise.

# Calculation: {exp} = {ans}

If the calculation is correct, return True. If not, return False.

Think step-by-step, and MUST output True or False at the end of your verification.

---

To use the tool, we prompt LM to generate a code instead of verification in natural language using Prompt A.2.

---

**Prompt A.2: Check Calculation with Code**

Generate a simple Python script that evaluates the correctness of a given mathematical calculation.

# Calculation: {exp} = {ans}
The script should print 'The calculation is correct' if the calculation is correct, otherwise print 'The calculation is incorrect'.

### Constraints:
- The output must be a single Python code block without any function definition.
- The script should evaluate the expression as a boolean comparison.

If the evaluated result of 'exp' matches 'ans', print 'The calculation is correct', otherwise print 'The calculation is incorrect'.

---

# B DETAILS OF OUR METHOD

The choice of the component of the tool-based verification function depends on the task:

1. **Mathematical reasoning task:** sLM generates executable programming code $c_1$, then the code interpreter executes the code and validates the correctness of computations (Schick et al., 2023; Gou et al., 2024b). Since the code interpreter executes the code as well as outputs the verification score, we can regard the extraction of the verification score part as the identity function, i.e., $c_2 = \mathcal{T}(c)$. Therefore, $f(\boldsymbol{x}, \boldsymbol{y}; \mathcal{T}, \theta)$ for numerical reasoning tasks can be represented as:
$$f(\boldsymbol{x}, \boldsymbol{y}; \mathcal{T}, \theta) = c_2 = \mathcal{T}(c_1), \quad \text{where} \quad c_1 \sim \pi(c \mid \boldsymbol{x}, \boldsymbol{y}, \boldsymbol{I}_c; \theta). \tag{8}$$

2. **Knowledge-intensive task:** The tool $\mathcal{T}$ acts as a retriever that returns a set of relevant knowledge passages $\boldsymbol{k}$ based on the input $\boldsymbol{x}$ and candidate response $\boldsymbol{y}$. Subsequently, sLM verifies the consistency between the retrieved knowledge $\boldsymbol{k}$ and the claims within $\boldsymbol{y}$. Since the retriever utilizes $\boldsymbol{x}$ and $\boldsymbol{y}$ directly as a query, we can regard the tool-calling query part as the identity function, i.e., $c_1 = (\boldsymbol{x}, \boldsymbol{y})$. Therefore, $f(\boldsymbol{x}, \boldsymbol{y}; \mathcal{T}, \theta)$ for knowledge-intensive tasks can be represented as:
$$f(\boldsymbol{x}, \boldsymbol{y}; \mathcal{T}, \theta) = c_2 \sim \pi\left(c \mid \mathcal{T}(c_1), \boldsymbol{x}, \boldsymbol{y}, \boldsymbol{I}_f; \theta\right), \quad \text{where} \quad c_1 = (\boldsymbol{x}, \boldsymbol{y}), \tag{9}$$

## C PROOF FOR THEORETICAL ANALYSIS

### C.1 FROM THEORETICAL ANALYSIS TO PRACTICE.

Our theoretical analysis illustrates two key ideas: (1) tool integration reduces the memorization burden of small language models, and (2) the two stage design improves test time scaling by enabling a more reliable filtering function. These results are based on simplified and idealized settings and are not intended to capture full practical behavior. Instead, they provide conceptual illustrations grounded in existing theoretical frameworks (Brown et al., 2021; Beirami et al., 2024). The empirical trends observed in our experiments are consistent with these theoretical intuitions, suggesting that the underlying principles extend to more complex real-world scenarios.

### C.2 PROOF OF LEMMA 5.1

*Proof.* $\mathcal{X}$ has cardinality $|\mathcal{X}| = 2 \cdot (M-1)^3 = \Theta(M^3)$. Also, Theorem 1.1 in Brown et al. (2021) says that any learning algorithms $\mathcal{A}$ that is $\varepsilon$-close-to-optimal with sufficiently small $\varepsilon > 0$ also satisfies that the mutual information between data samples and the model learned by $\mathcal{A}$ given the data distribution is proportional to at least the number of data samples multiplying dimension of data dimension. Since the number of data samples' cardinality in subsection 5.1 is $\Theta(M^3)$ and the data dimension is 1, we can state that $I(X; \theta \mid P) = \Omega(M^3)$. This proof says that if a model directly memorizes which $(a, b, c)$ pairs map to each $c = a + b$ with near-zero error, $\theta$ must encode on the order of $M^3$ bits of information about $X$. $\square$

### C.3 PROOF OF THEOREM 5.2

*Proof.* In this case, $\mathcal{A}$ has access to the tool function as follows:

$$f(a, b, c; \mathcal{T}) = \mathbf{1}_{a+b=c}.$$

Then, $\theta = \mathcal{A}(X)$ such that $f(a, b, c; \theta, \mathcal{T}) = \mathbf{1}_{a+b=c}$. Regardless of $X$, $f$ can perfectly get label $r$, resulting in $\mathrm{err}_{q,|\mathcal{X}|}(\mathcal{A}) = 0$.

In addition, since $\theta$ is determined regardless of $X$ through $\mathcal{A}$, we get

$$
\begin{aligned}
I(X; \theta \mid P) &= H(\theta \mid P) - H(\theta \mid X, P) \\
&= H(\theta \mid P) - H(\theta \mid P) \\
&= 0,
\end{aligned}
$$

where $H$ is the entropy. Therefore, we can state that $I(X; \theta \mid P) = 0$. $\square$

### C.4 PROOF OF THEOREM 5.3

We first show Best-of-$N$ accuracy first in Lemma C.1 . Then we show the monotonicity with respect to the $p$.

**Lemma C.1** (Best-of-$N$ Accuracy with Imperfect Verifier). *Let the generator output 0 or 1 with equal probability, i.e.,*

$$\pi(0|\boldsymbol{x}) = \pi(1|\boldsymbol{x}) = \frac{1}{2},$$

*and let the verifier $r$ with noise level $p, q$ be defined as follows:*

$$
r_{p,q}(\boldsymbol{x}, 0) = \begin{cases} 0, & \text{w.p. } p, \\ 1, & \text{w.p. } 1-p, \end{cases} \qquad r_{p,q}(\boldsymbol{x}, 1) = \begin{cases} 1, & \text{w.p. } q, \\ 0, & \text{w.p. } 1-q, \end{cases}
$$

*with the condition that $q > 1 - q$ and $p > 1 - p$. Then, for $N \geq 1$, the probability that the best-of-$N$ output is ground truth label, i.e. 1, is given by*

$$\pi^N(1 \mid \boldsymbol{x}) = \frac{q}{q+1-p}\left[1 - \left(\frac{1-q+p}{2}\right)^N\right] + \frac{1-q}{1-q+p}\left(\frac{1-q+p}{2}\right)^N. \tag{10}$$

*Proof.* A single sample from the generator $\pi$ is labeled 1 with probability $\frac{1}{2}$ and 0 with probability $\frac{1}{2}$. Given the verifier $r$ with noise level $p$ and $q$, the joint probability are:

$$
\begin{aligned}
P\left(y=1, r=1 \mid \boldsymbol{x}\right) &= \frac{1}{2} q, \\
P\left(y=0, r=1 \mid \boldsymbol{x}\right) &= \frac{1}{2} (1-p), \\
P\left(y=1, r=0 \mid \boldsymbol{x}\right) &= \frac{1}{2} (1-q), \\
P\left(y=0, r=0 \mid \boldsymbol{x}\right) &= \frac{1}{2} p.
\end{aligned}
\tag{11}
$$

Then, the probability that a single sample yields a verifier score of 1 and 0 are

$$
p\left(r=1 \mid \boldsymbol{x}\right) = \sum_{i \in \{0,1\}} p\left(y=i, r=1 \mid \boldsymbol{x}\right) = \frac{1}{2} (1-p) + \frac{1}{2} (q),
$$

$$
p\left(r=0 \mid \boldsymbol{x}\right) = 1 - p\left(r=1 \mid \boldsymbol{x}\right) = \frac{1+p-q}{2},
$$

respectively.

Define the event

$$
A = \left\{ \text{at least one Best-of-}N \text{ sample is } r(\mathbf{x}, y)=1 \right\}.
$$

Then, from Equation 11, the probability that all $N$ candidates yield $r=0$ is

$$
P(A^c) = P(r=0)^N = \left( \frac{1+p-q}{2} \right)^N,
\tag{12}
$$

and consequently,

$$
P(A) = 1 - P(A^c) = 1 - \left( \frac{1+p-q}{2} \right)^N.
\tag{13}
$$

Consider two cases:

**Case 1.** *At least one candidate yields $r=1$ (event $A$ occurs).* In this case, the final output is chosen uniformly among the candidates with $r=1$. For any candidate with $r=1$, the probability that it originated from $y=1$ is given by

$$
P\left(y=1 \mid r=1, \boldsymbol{x}\right) = \frac{P(y=1, r=1 \mid \boldsymbol{x})}{P(r=1 \mid \boldsymbol{x})} = \frac{\frac{1}{2}q}{\frac{1}{2}(q+1-p)} = \frac{q}{q+1-p}.
$$

**Case 2.** *All candidates yield $r=0$ (event $A^c$ occurs).* In this case, the output is chosen uniformly among all $N$ candidates. For a candidate with $r=0$, the probability that it is 1 is

$$
P\left(y=1 \mid r=0, \boldsymbol{x}\right) = \frac{P(y=1, r=0 \mid \boldsymbol{x})}{P(r=0 \mid \boldsymbol{x})} = \frac{\frac{1}{2}(1-q)}{\frac{1}{2}(p+1-q)} = \frac{1-q}{p+1-q}.
$$

By the law of total probability, the overall probability that the Best-of-$N$ output is 1 is

$$
\begin{aligned}
\pi^N\left(1 \mid \mathbf{x}\right) &= P(A) \cdot P\left(y=1 \mid r=1, \boldsymbol{x}\right) + P(A^c) \cdot P\left(y=1 \mid r=0, \boldsymbol{x}\right) \\
&= P(A) \cdot \frac{q}{q+1-p} + P(A^c) \cdot \frac{1-q}{p+1-q}.
\end{aligned}
$$

From Equation 12 and Equation 13, we have

$$
\pi^N\left(1 \mid \mathbf{x}\right) = \left( 1 - \left( \frac{1+p-q}{2} \right)^N \right) \frac{q}{q+1-p} + \left( \frac{1+p-q}{2} \right)^N \frac{1-q}{p+1-q}.
\tag{14}
$$

$\square$

Using Lemma C.1 , Theorem 5.3 is proven as follows:

*Proof.* Define the difference $\Delta = p - q$, so that

$$\frac{1 + p - q}{2} = \frac{1 + \Delta}{2},$$

and note that

$$q + 1 - p = 1 - \Delta \quad \text{and} \quad p + 1 - q = 1 + \Delta.$$

Then, from Lemma C.1 , Equation 14 is rewritten as

$$f(\Delta) = \left(1 - \left(\frac{1 + \Delta}{2}\right)^N\right) \frac{q}{1 - \Delta} + \left(\frac{1 + \Delta}{2}\right)^N \frac{1 - q}{1 + \Delta}. \tag{15}$$

Since $q$ is held fixed, an increase in $p$ corresponds to an increase in $\Delta$.

Define

$$A(\Delta) = \left(\frac{1 + \Delta}{2}\right)^N,$$

so that

$$f(\Delta) = [1 - A(\Delta)] \frac{q}{1 - \Delta} + A(\Delta) \frac{1 - q}{1 + \Delta}.$$

Differentiating equation 15 with respect to $\Delta$ yields

$$f'(\Delta) = -A'(\Delta)\frac{q}{1 - \Delta} + [1 - A(\Delta)] \frac{q}{(1 - \Delta)^2} + A'(\Delta)\frac{1 - q}{1 + \Delta} - A(\Delta)\frac{1 - q}{(1 + \Delta)^2}, \tag{16}$$

with

$$A'(\Delta) = \frac{N}{2} \left(\frac{1 + \Delta}{2}\right)^{N-1}.$$

Reformulating Equation 16 results in

$$f'(\Delta) = \frac{q}{(1 - \Delta)^2} \underbrace{\left[1 - A(\Delta) - (1 - \Delta) A'(\Delta)\right]}_{(a)} + \frac{1 - q}{(1 + \Delta)^2} \underbrace{\left[(1 + \Delta) A'(\Delta) - A(\Delta)\right]}_{(b)}.$$

(a) Define $x = \frac{1+\Delta}{2}$. Then,

$$1 - A(\Delta) - (1 - \Delta) A'(\Delta) = 1 - x^N - N(1 - x) x^{N-1}.$$

Set $g(x) = 1 - x^N - N(1 - x) x^{N-1}$. Then, $g(0) = 1$, $g(1) = 0$, and $g'(x) \leq 0$ induces $g(x) >= 0$ for $x \in [0, 1]$. Since $q1 - q2 \in [-1, 1]$, $x \in [0, 1]$ is satisfied. Therefore, we have

$$1 - A(\Delta) - (1 - \Delta) A'(\Delta) > 0.$$

(b) Since $A'(\Delta) = \frac{N}{2} \left(\frac{1+\Delta}{2}\right)^{N-1}$ and $A(\Delta) = \left(\frac{1+\Delta}{2}\right)^N$, we have

$$(1 + \Delta) A'(\Delta) = (1 + \Delta) \frac{N}{2} \left(\frac{1 + \Delta}{2}\right)^{N-1}$$

$$= N \left(\frac{1 + \Delta}{2}\right)^N$$

$$= N A(\Delta).$$

Hence, we have

$$(1 + \Delta) A'(\Delta) - A(\Delta) = N A(\Delta) - A(\Delta)$$
$$= (N - 1) A(\Delta).$$

For $N \geq 2$ and $\Delta > -1$, both $N - 1 > 0$ and $A(\Delta) > 0$. Therefore,

$$(1 + \Delta) A'(\Delta) - A(\Delta) > 0.$$

Since (a) and (b) are non-negative, we conclude that $f'(\Delta) > 0$ for all $\Delta$ and $N \geq 2$.

Thus, for $N \geq 2$, if $\bar{p} > \underline{p}$ (i.e. $\Delta^1 > \Delta^2$), it follows that

$$f(\Delta^1) > f(\Delta^2),$$

or equivalently,

$$\pi^N(1 \mid \mathbf{x})\Big|_{p=\bar{p}} > \pi^N(1 \mid \mathbf{x})\Big|_{p=\underline{p}}.$$

$\square$

## D   IMPLEMENTATION DETAILS

### D.1   MODEL

We use LLaMA-3.2-1B-Instruct (Dubey et al., 2024), Qwen-2.5-0.5B-Instruct (Yang et al., 2024), SmolLM2-360M-Instruct (Allal et al., 2025) as base models for our experiments.

### D.2   TRAINING

**Hyperparameters & Setting**   As mentioned in subsection 4.2, we fine-tune small language models (sLMs) for each module using LoRA (Hu et al., 2022). However, for PRM, we fine-tune the full model including the classifier head following Wang et al. (2024a). Only for SmolLM2-360M-Instruct, we fine-tune the model on generation as it achieves under $10\%$ accuracy on both GSM8K and MATH. We organize the hyperparameter details in Table 2. We use 4 A100 40GB GPUs with FSDP (Zhao et al., 2023) for training.

Table 2: Hyperparameters used in fine-tuning sLM for each component.

| Hyperparameter | Verifier | PRM | ToolV |
|---|---|---|---|
| Learning rate | $1 \times 10^{-4}$ | $1 \times 10^{-5}$ | $1 \times 10^{-4}$ |
| Batch size | 16 | 16 | 16 |
| Max length | 2048 | 2048 | 2048 |
| LoRA rank | 64 | - | 64 |
| LoRA $\alpha$ | 128 | - | 128 |
| Optimizer | AdamW | AdamW | AdamW |
| Training epochs | 1 | 3 | 3 |
| Scheduler | Linear | Linear | Linear |

**Dataset for Distillation**   We perform distillation using the training split of each dataset. For MMLU-Pro, we adopt the train-test split provided by Zeng et al. (2025). Training dataset sizes are 7473 for GSM8K, 7500 for MATH, 1284 for MMLU-Pro.

During distillation, we prompt the teacher model—gpt-4o-mini-2024-07-18 in our experiments—to generate sequences used as supervision for training. For the generative verifier, we follow the prompt design from Zhang et al. (2024). Specifically, we generate 8 completions per problem using a temperature of 0.6, and treat these outputs as the training data.

For code generation tasks, we apply the Prompt D.1. Using this prompt, we generate 4 completions per problem at a temperature of 0.6, which are then used as training samples. For fact-checking in MMLU-Pro, we similarly generate 8 completions per problem with a temperature of 0.6, using the teacher model to construct the training dataset. We use the Prompt D.2. In addition, we retrieve 3 documents for fact-checking from wikipedia abstracts using BM25 implemtented in Pyserini (Lin et al., 2021).

---

**Prompt D.1: Code Generation**

```
SYSTEM_PROMPT:
```
Write a Python code block that verifies whether a given solution is correct based on the provided question, following these guidelines:

- The code should be a single Python block, formatted as:
"'python
CODE
"'

- The code should only print True if the solution is verified as correct. Otherwise, it should only print False if the solution is incorrect.
- Use only the following built-in modules where necessary:
    - 'math' (for floating-point comparisons using math.isclose())
    - 'sympy' (for symbolic calculations, including $\pi$ and fractions)
    - 'cmath' (for complex number operations)
- For floating-point comparisons, use math.isclose() instead of ==.
- Use 'sympy.pi' for $\pi$ and 'sympy.Rational' for fractions.
- Simplify all fractions and square roots without converting them to decimal values.

```
USER_PROMPT:
```
### Input
- Question: {question}
- Solution: {solution}

### Output:
Return python code only.

---

**Prompt D.2: Fact-checking generation**

```
SYSTEM_PROMPT:
```
You are a domain expert.

```
USER_PROMPT:
```
Check the factual correctness of each statement in the provided solution to the question, using only the information available in the given document.
- Evaluate only the explicit factual claims made in the solution. Do not verify or evaluate the final conclusion or answer itself (e.g., The answer is ...).
- If a statement is factually incorrect based on the document, mark it as incorrect.
- If a statement cannot be verified using the document (i.e., the document does not confirm or deny it), treat it as not verifiable, and assume it is correct for the purpose of final verification.

⟨question⟩{question}⟨/question⟩

⟨document⟩{document}⟨/document⟩

⟨solution⟩{solution}⟨/solution⟩
At the end of the fact check, provide a final summary in the following format: Verification: Are all statements correct? (Yes/No)? X (where X is either Yes or No).
If any verifiably false statement is found, output: Verification: Are all statements correct? (Yes/No)? No
If no false statements are found (i.e., all are either correct or unverifiable), output: Verification: Are all statements correct? (Yes/No)? Yes

Table 3: Performance comparison between GenRM and ToolV + GenRM. Results are from experiments with Llama-3.2-1B-Instruct on MATH500 benchmark.

| Method | Accuracy | Precision | Recall | F1 Score |
|---|---|---|---|---|
| GenRM | 80.91% | 0.6153 | **0.7759** | 0.6863 |
| GenRM + ToolV | **86.99%** | **0.7666** | 0.7427 | **0.7545** |

Table 4: Performance of LLama-3.2-1B-Instruct on the MATH500 benchmark for Python code generation, using teacher model outputs as reference (gold). We set rejection as positive label for computing precision, recall, and f1 score.

| Model Size | Accuracy | Precision | Recall | F1 Score |
|---|---|---|---|---|
| 1B | 0.7687 | 0.8720 | 0.6946 | 0.7733 |
| 3B | **0.7973** | 0.8949 | **0.7286** | **0.8033** |
| 8B | 0.7906 | **0.9207** | 0.6908 | 0.7893 |

## D.3 EVALUATION

**Hyperparameters & Setting** We generate $N = 64$ solutions using a temperature of 0.8. In the case of GenRM, we follow the chain-of-thought variant proposed by Zhang et al. (2024). As in their setup, we generate $n = 8$ rationales and average the correctness scores across them, following the self-consistency method (Wang et al., 2023), using a temperature of 0.6. For PRM, we apply the final score aggregation approach, consistent with previous studies (Wang et al., 2024a; Snell et al., 2024). When using ToolV for mathematical reasoning, we generate 4 code completions with a temperature of 0.6 and consider the result correct if at least one of the generated codes passes. In knowledge-intensive reasoning with ToolV, we generate 4 rationales at the same temperature and consider the result correct only if all of them pass. In the case of MMLU-Pro, we retrieve three documents following the training setup. Gold documents are generated from each question using GPT-4o.

## E ADDITIONAL EXPERIMENTAL RESULTS

### E.1 TOOLV ON MULTI-DOMAIN KNOWLEDGE-INTENSIVE TASKS

We demonstrate that ToolV is effective in verifying solutions across a range of knowledge-intensive reasoning tasks from the subset of MMLU-Pro benchmark (Health, Economics, History domains) (Wang et al., 2024b). We adapt ToolV to function as a *fact-checker*, verifying claims in solutions without other components such as query transformation and reranker (Wei et al., 2024; Kang et al., 2023). We provide experimental results on the MMLU-Pro benchmark with minimal framework in this work (Wang et al., 2024b).

As shown in Figure 12, ToolV outperforms the distilled PRM baseline derived from the VersaPRM (Zeng et al., 2025). For the tool, we retrieve three documents from Wikipedia using BM25. Due to variability in document quality, performance is somewhat unstable in some cases. To explore ToolV's upper bound, we also evaluate it using gold documents generated by GPT-4o. Results show that ToolV performance improves significantly with higher-quality documents, demonstrating its potential for multi-domain knowledge-intensive reasoning.

In Figure 13, we plot the best-of-N results for all N values used in the experiments from Figure 12. Compared to math reasoning tasks, the plot is less clearly separated. However, ToolV + PRM generally outperforms the other baselines and clearly surpasses them even when using gold documents.

### E.2 ACCURACY OF GENERATIVE REWARD MODEL IN VERIFICATION GENERATION

In Table 3, we report the accuracy of GenRM with and without ToolV. The results show that ToolV significantly improves accuracy, precision, and F1 score, indicating that it effectively removes false positive cases among the solutions. The confusion matrix in Figure 14 further illustrates this trend.

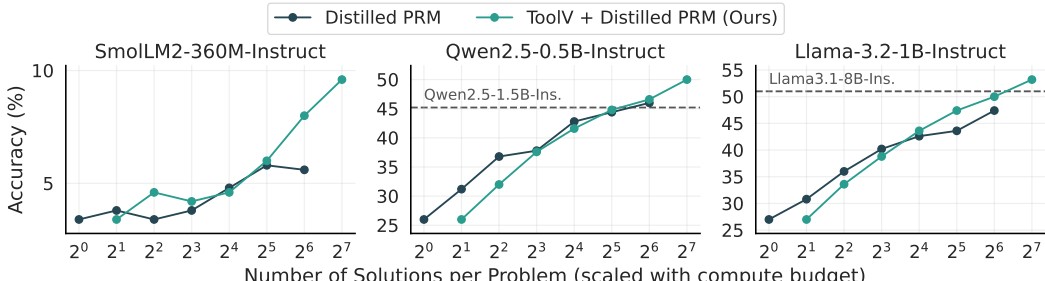

Figure 11: **MATH500 with PRM under a scaled x-axis reflecting compute budget.** PRM only performs better at small budgets, but our method surpasses it as test-time scaling increases.

However, it also reveals that ToolV occasionally removes true positives, primarily due to incorrectly generated Python code.

### E.3    ACCURACY OF TOOL-BASED VERIFIER IN CODE GENERATION

In Table 4, we present the accuracy of Python code generation, treating the teacher-generated code as the ground truth. The results show that precision is quite high—even for the 1B model, the distilled 1B ToolV is able to filter out more than 85% of incorrect solutions among the incorrect solutions that teacher model predicted. However, the recall is low, indicating that the generated code sometimes mistakenly filters out correct solutions.

### E.4    DATA EFFICIENCY OF RM AND TOOL-BASED VERIFIER IN DISTILLATION

In Figure 15, we conduct experiments to evaluate how much data is needed for each RM-based and tool-based verifier to achieve satisfactory performance. In each plot, we reduce the distillation data to 10% and 1% for one verifier, while keeping the other verifier fully distilled using 100% of the dataset. The results show that ToolV maintains competitive performance even with only 10% of the data, demonstrating its data efficiency during distillation.

### E.5    NECESSITY OF TWO-STAGE DESIGN

A natural question is whether ToolV alone is sufficient for verification with small models. To answer this, we conduct ablation studies that isolate the contribution of each stage. We report results in Table 6 and Table 7.

ToolV alone is a strong verifier. It already surpasses PRM on MATH500 and GenRM on GSM8K. This shows that executable consistency checks provide an effective signal for filtering incorrect solutions. However, ToolV focuses primarily on execution-based correctness such as calculation. It does not capture higher-level reasoning errors that are not executable, such as misinterpreting problem structure or producing logically inconsistent intermediate steps.

Our ablation results show that this complementary relationship leads to consistent performance gains. ToolV alone improves over PRM and GenRM, but the two stage design (ToolV plus PRM or GenRM) achieves the highest accuracy across both MATH500 and GSM8K. These findings confirm that the two stages provide different verification signals and that both are necessary for strong verification under test time scaling with small models.

### E.6    COMPUTE BUDGET DISCUSSION WITH PRM

We extend the discussion in Section 6.4. Since PRMs require no generation, adding ToolV can be interpreted as allocating roughly twice the compute budget compared to using PRM alone. Figure 11 presents the compute scaled comparison. PRM without ToolV performs better when the number of generated solutions is small, where the additional verification cost is not yet leveraged. As the compute budget increases, ToolV combined with PRM consistently achieves higher accuracy than PRM alone. This shows that ToolV remains effective under test-time scaling, especially for small model settings.

Table 5: Performance comparison of GenRM variants across different numbers of generated solutions $N$.

| Method | $N = 1$ | $N = 2$ | $N = 4$ | $N = 8$ | $N = 16$ | $N = 32$ | $N = 64$ |
|---|---|---|---|---|---|---|---|
| One-stage GenRM | 27.0 | 31.6 | 36.0 | 40.0 | 42.0 | 44.8 | 45.4 |
| Two-stage GenRM | 27.0 | 29.6 | 35.0 | 39.8 | 42.0 | 44.4 | 46.0 |
| GenRM + ToolV | 27.0 | **33.8** | **38.8** | **43.8** | **47.2** | **49.4** | **50.6** |

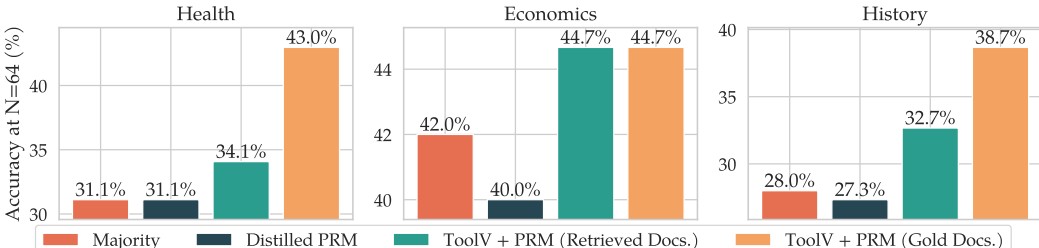

Figure 12: **MMLU-Pro with PRM.** Weighted Best-of-N ($N = 64$) performance of Llama-3.2-1B-Instruct on three knowledge-intensive domains, illustrating the effect of different document sources in ToolV + Distilled PRM (retrieved and gold documents). ToolV extends beyond math, improving PRM on multi-domain knowledge-intensive reasoning tasks.

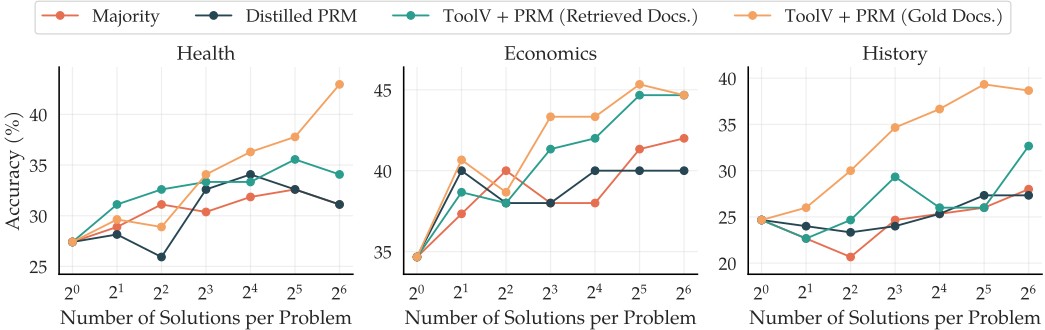

Figure 13: **MMLU-Pro with PRM (Line Plot).** Weighted Best-of-N performance of Llama-3.2-1B-Instruct on three knowledge-intensive domains from MMLU-Pro.

### E.7 TWO-STAGE GENRM WITHOUT TOOL INTEGRATION

Table 5 reports the performance of two stage verification using GenRM without any tool integration. The results show that simply adding an additional verification stage does not provide the gains observed with ToolV. This confirms that the improvements come from tool integration itself rather than from increased compute through multi-stage verification.

### E.8 EXACT PERFORMANCE OF THE PLOT

To provide clear measurements corresponding to Figure 3, Figure 4, and Figure 5, we include tables that report the exact performance values for each model.

For Figure 3, the exact scores are shown in Table 8, Table 9, and Table 10 for SmolLM2-360M-Instruct, Qwen2.5-0.5B-Instruct, and Llama-3.2-1B-Instruct, respectively.

For Figure 4, the corresponding tables are provided in Table 11, Table 12, and Table 13.

For Figure 5, detailed results appear in Table 14, Table 15, and Table 16.

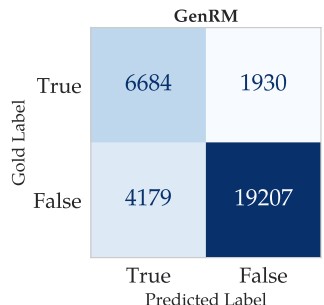
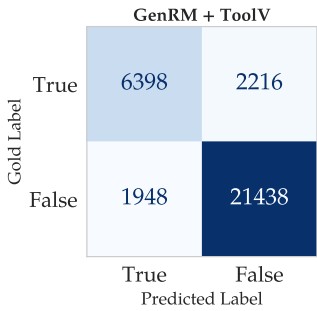

Figure 14: Confusion matrix of verification results from GenRM and GenRM + ToolV, where True denotes the correct solution. This result indicates ToolV improves the performance on removing false positive cases. Results are from experiments with Llama-3.2-1B-Instruct on MATH500 benchmark.

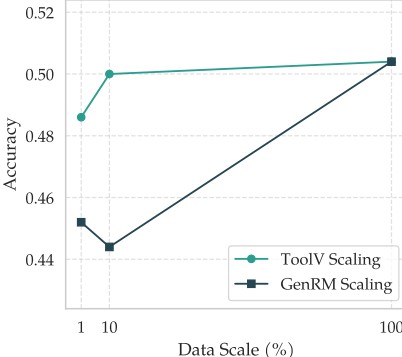

Figure 15: **Data-scale experiment.** Performance comparison with varying distillation data sizes. In each plot, one verifier is distilled with 10% or 1% of data, while the other uses the full dataset. ToolV remains competitive even with only 10% of data, highlighting its data efficiency. Results are from experiments with Llama-3.2-1B-Instruct on MATH500.

## F  CASE ANALYSIS

In this section, we present cases where **ToolV** either enables or fails the self-verification process. All examples are taken from level 4 problems in **MATH500**.

In Example G.1, the solution is incorrect—it computes $37 \times 2$ as $374$, which is wrong. However, the verifier (**GenRM-CoT**) fails to detect this error and incorrectly marks the step as correct, ultimately concluding that the entire solution is valid. In this case, **ToolV** implements a program that checks the correctness of the solution by solving the equation using the `SymPy` library. As expected, the result of the check is `False`, successfully identifying and removing the incorrect solution.

Occasionally, **ToolV** may produce incorrect Python code, as discussed in subsection E.3. To illustrate such a case, we present a failure example in Example G.2. Here, the solution is actually correct, and **GenRM** correctly verifies it. However, **ToolV** generates code that leads to an incorrect conclusion. Although the code appears reasonable, the comparison `sp.simplify(result) == sp.simplify(expected_result)` might return `False`, since symbolic expressions can differ in form even when they are mathematically equivalent. To properly compare equations, the code should instead use `sp.Eq(...)` which is more reliable for symbolic equality checks.

## G  LLM USAGE

We used large language models (LLMs) solely as a writing assistant, for improving grammar and clarity of the paper. No part of the research ideation, experimental design, or analysis relied on LLMs.

| MATH500 | Accuracy |
|---|---|
| PRM | 47.4 |
| ToolV | 52.4 |
| ToolV + PRM | 53.2 |

Table 6: Ablation on ToolV with PRM verifier with Llama-3.2-1B-Instruct.

| GSM8K | Accuracy |
|---|---|
| GenRM | 71.87 |
| ToolV | 73.62 |
| ToolV + GenRM | 74.60 |

Table 7: Ablation on ToolV with GenRM verifier with Llama-3.2-1B-Instruct.

Table 8: Weighted Best-of-N accuracy on MATH500 for the SmolLM2-360M-Instruct with PRM verification. ToolV provides clear improvements over PRM and majority voting.

| Method | 1 | 2 | 4 | 8 | 16 | 32 | 64 |
|---|---|---|---|---|---|---|---|
| Majority | 3.4 | 3.4 | 3.4 | 3.8 | 5.0 | 5.8 | 6.2 |
| PRM Verifier | 3.4 | 3.8 | 3.4 | 3.8 | 4.8 | 5.8 | 5.6 |
| ToolV + PRM Verifier (Ours) | 3.4 | 4.6 | 4.2 | 4.6 | 6.0 | 8.0 | 9.6 |

Table 9: Weighted Best-of-N accuracy on MATH500 for the Qwen2.5-0.5B-Instruct model with PRM verification. ToolV consistently improves over PRM verification and majority voting.

| Method | 1 | 2 | 4 | 8 | 16 | 32 | 64 |
|---|---|---|---|---|---|---|---|
| Majority | 26.0 | 26.0 | 32.2 | 37.0 | 39.2 | 41.6 | 42.8 |
| PRM Verifier | 26.0 | 31.2 | 36.8 | 37.8 | 42.8 | 44.4 | 46.0 |
| ToolV + PRM Verifier (Ours) | 26.0 | 32.0 | 37.6 | 41.6 | 44.8 | 46.6 | 50.0 |

Table 10: Weighted Best-of-N accuracy on MATH500 for the Llama-3.2-1B-Instruct model with PRM verification. ToolV provides clear gains over PRM based verification and majority voting methods.

| Method | 1 | 2 | 4 | 8 | 16 | 32 | 64 |
|---|---|---|---|---|---|---|---|
| Majority | 27.0 | 27.0 | 33.2 | 38.8 | 41.6 | 43.8 | 45.8 |
| PRM Verifier | 27.0 | 30.8 | 36.0 | 40.2 | 42.6 | 43.6 | 47.4 |
| ToolV + PRM Verifier (Ours) | 27.0 | 33.6 | 38.8 | 43.6 | 47.4 | 50.0 | 53.2 |

Table 11: Weighted Best-of-N accuracy on MATH500 for the SmolLM2-360M-Instruct model with GenRM verification. ToolV provides consistent gains over verifier based and majority voting methods.

| Method | 1 | 2 | 4 | 8 | 16 | 32 | 64 |
|---|---|---|---|---|---|---|---|
| Majority | 3.2 | 3.2 | 3.2 | 3.6 | 4.8 | 5.6 | 5.8 |
| Zero shot GenRM | 3.2 | 3.2 | 3.2 | 3.6 | 4.8 | 5.6 | 5.8 |
| Distilled GenRM | 3.2 | 3.4 | 3.8 | 3.8 | 5.4 | 7.2 | 7.0 |
| ToolV + Distilled GenRM (Ours) | 3.2 | 4.0 | 4.0 | 5.2 | 6.8 | 8.4 | 10.8 |

Table 12: Weighted Best-of-N accuracy on MATH500 for the Qwen2.5-0.5B-Instruct model with GenRM verification. The results show that ToolV provides clear gains over verifier based and majority voting methods by filtering calculation errors that generative verification cannot resolve.

| Method | 1 | 2 | 4 | 8 | 16 | 32 | 64 |
|---|---|---|---|---|---|---|---|
| Majority | 26.0 | 26.0 | 32.2 | 37.0 | 39.2 | 41.6 | 42.8 |
| Zero shot GenRM | 26.0 | 27.6 | 31.0 | 37.4 | 39.8 | 42.6 | 43.0 |
| Distilled GenRM | 26.0 | 31.6 | 34.4 | 38.4 | 40.8 | 43.8 | 44.6 |
| ToolV + Distilled GenRM (Ours) | 26.0 | 32.0 | 36.8 | 40.8 | 44.0 | 45.0 | 47.2 |

Table 13: Weighted Best-of-N accuracy on MATH500 for the Llama-3.2-1B-Instruct model with GenRM verification. The results show that ToolV provides clear gains over verifier based and majority voting methods by filtering calculation errors that generative verification cannot resolve.

| Method | 1 | 2 | 4 | 8 | 16 | 32 | 64 |
|---|---|---|---|---|---|---|---|
| Majority | 27.0 | 27.0 | 33.2 | 38.8 | 41.6 | 43.8 | 45.8 |
| Zero-shot GenRM | 27.0 | 28.0 | 33.8 | 38.8 | 40.6 | 44.4 | 46.0 |
| Distilled GenRM | 27.0 | 31.6 | 36.0 | 40.0 | 42.0 | 44.8 | 45.4 |
| ToolV + Distilled GenRM (Ours) | 27.0 | 33.8 | 38.8 | 43.8 | 47.2 | 49.4 | 50.6 |

Table 14: Weighted Best-of-N accuracy on GSM8K for the SmolLM2-360M-Instruct model with GenRM verification. ToolV provides clear gains over verifier based and majority voting methods.

| Method | 1 | 2 | 4 | 8 | 16 | 32 | 64 |
|---|---|---|---|---|---|---|---|
| Majority | 10.92 | 10.92 | 13.57 | 18.35 | 23.43 | 26.69 | 28.51 |
| Zero shot GenRM | 10.92 | 10.92 | 13.57 | 18.35 | 23.43 | 26.69 | 28.51 |
| Distilled GenRM | 10.92 | 14.94 | 18.80 | 23.05 | 26.69 | 29.72 | 31.39 |
| ToolV + Distilled GenRM (Ours) | 10.92 | 15.31 | 20.92 | 26.54 | 32.75 | 36.85 | 39.88 |

Table 15: Weighted Best-of-N accuracy on GSM8K for the Qwen2.5-0.5B-Instruct model with GenRM verification. ToolV consistently outperforms verifier based and majority voting methods.

| Method | 1 | 2 | 4 | 8 | 16 | 32 | 64 |
|---|---|---|---|---|---|---|---|
| Majority | 44.28 | 44.28 | 50.49 | 56.10 | 59.14 | 60.80 | 61.71 |
| Zero shot GenRM | 44.28 | 43.44 | 50.11 | 56.63 | 59.82 | 61.18 | 62.62 |
| Distilled GenRM | 44.28 | 49.89 | 55.65 | 59.59 | 62.70 | 64.52 | 66.03 |
| ToolV + Distilled GenRM (Ours) | 44.28 | 50.87 | 56.56 | 62.62 | 64.90 | 66.57 | 68.31 |

Table 16: Weighted Best-of-N accuracy on GSM8K for the Llama-3.2-1B-Instruct model with GenRM verification. ToolV shows consistent improvements over verifier based and majority voting methods.

| Method | 1 | 2 | 4 | 8 | 16 | 32 | 64 |
|---|---|---|---|---|---|---|---|
| Majority | 49.96 | 49.96 | 58.15 | 64.44 | 67.17 | 68.46 | 69.45 |
| Zero shot GenRM | 49.96 | 51.18 | 58.76 | 64.75 | 67.48 | 68.99 | 70.58 |
| Distilled GenRM | 49.96 | 57.09 | 61.94 | 66.94 | 69.67 | 71.11 | 71.87 |
| ToolV + Distilled GenRM (Ours) | 49.96 | 58.15 | 64.29 | 68.23 | 71.04 | 73.62 | 74.60 |

---

**Example G.1: Success case 1 of ToolV**

**Problem:** Half the value of $3x - 9$ is $x + 37$. What is the value of $x$?

---

**Solution:** ## Step 1: Translate the given information into an equation
We are given that half the value of $3x - 9$ is $x + 37$. This can be translated into the equation $\frac{3x-9}{2} = x + 37$.
## Step 2: Multiply both sides of the equation by 2 to clear the fraction
Multiplying both sides of the equation by 2 to clear the fraction gives us $3x - 9 = 2x + 374$.
## Step 3: Subtract $2x$ from both sides of the equation to isolate the variable term on one side
Subtracting $2x$ from both sides of the equation gives us $x - 9 = 374$.
## Step 4: Add 9 to both sides of the equation to further isolate the variable term
Adding 9 to both sides of the equation gives us $x = 383$.
The final answer is: $\boxed{383}$

---

**GenRM-CoT:** Let's verify the provided solution step by step.
## Step 1: Translate the given information into an equation
The solution states that half the value of $3x - 9$ is $x + 37$. The translation into the equation $\frac{3x-9}{2} = x + 37$ is correct.
**Verification:** This step is correct.
## Step 2: Multiply both sides of the equation by 2 to clear the fraction
The solution claims that multiplying both sides by 2 gives $3x - 9 = 2x + 374$. **This is accurate since multiplying both sides by 2 eliminates the fraction.**
**Verification:** This step is correct.
## Step 3: Subtract $2x$ from both sides of the equation to isolate the variable term on one side
The solution states that subtracting $2x$ from both sides results in $x - 9 = 374$. This is also correct.
**Verification:** This step is correct.
## Step 4: Add 9 to both sides of the equation to further isolate the variable term
The solution states that adding 9 to both sides gives $x = 383$. This is accurate as well, as adding 9 to $x - 9$ gives $x + 9 = 374$ and thus $x = 374 - 9$.
**Verification:** This step is correct.
The final answer is given as $\boxed{383}$.
### Final Verification
All steps have been verified to be correct, and the final answer is consistent with the calculations.
Verification: Is the answer correct (Yes/No)? Yes

---

**ToolV**: `import sympy as sp`

```
# Define the variable
x = sp.symbols('x')

# Given equation
equation = sp.Eq((3*x - 9) / 2, x + 37)

# Solve the equation
solution = sp.solve(equation, x)[0]

# Verify the solution
is_correct = sp.simplify(solution) == 383
print(is_correct)
```
**Interpreter output**: False

---

**Example G.2: Failure case 1 of ToolV**

---

**Problem:** The operation $\&$ is defined for positive integers $a$ and $b$ as $a\&b = \dfrac{\sqrt{ab + a}}{\sqrt{ab - b}}$. What is the value of $9\&2$? Express your answer as a common fraction in simplest radical form.

---

**Solution:** ## Step 1: Substitute the given values of $a$ and $b$ into the definition of $\&$. We have $a = 9$ and $b = 2$, so we can substitute these values into the definition of $\&$ as follows:

$$9 \& 2 = \frac{\sqrt{(9)(2) + 9}}{\sqrt{(9)(2) - 2}}$$

## Step 2: Simplify the expressions inside the square roots. We can simplify the expressions inside the square roots as follows:

$$9 \& 2 = \frac{\sqrt{18 + 9}}{\sqrt{18 - 2}} = \frac{\sqrt{27}}{\sqrt{16}}$$

## Step 3: Further simplify the expressions by finding square roots of perfect squares inside the fraction. We can simplify the expressions by finding square roots of perfect squares inside the fraction as follows:

$$9 \& 2 = \frac{3\sqrt{3}}{4}$$

The final answer is: $\boxed{\dfrac{3\sqrt{3}}{4}}$

---

**GenRM-CoT:** Let's verify the solution step by step.
## Step 1: Substituting values The solution correctly substitutes $a = 9$ and $b = 2$ into the expression for $9 \& 2$:

$$9 \& 2 = \frac{\sqrt{(9)(2) + 9}}{\sqrt{(9)(2) - 2}}.$$

This step is correct.
## Step 2: Simplifying inside the square roots The next part of the solution simplifies the expressions inside the square roots:

$$9 \& 2 = \frac{\sqrt{18 + 9}}{\sqrt{18 - 2}} = \frac{\sqrt{27}}{\sqrt{16}}.$$

This simplification is also correct.
## Step 3: Finding square roots The solution proceeds to find the square roots:

$$\frac{3\sqrt{3}}{4}.$$

This simplification is correct as $\sqrt{27} = 3\sqrt{3}$ and $\sqrt{16} = 4$.
## Final Answer The final answer given is:

$$\boxed{\dfrac{3\sqrt{3}}{4}}.$$

This matches the expected answer provided in the verification section.
In summary, each step in the solution was executed correctly and led to the correct final answer.
Verification: Is the answer correct (Yes/No)? **Yes**

---

## Example G.3: Failure case 1 of ToolV (cont.)

**ToolV:** `import sympy as sp`

# Define the operation
```
def operation(a, b):
    return sp.sqrt((a * b + a) / (a * b - b))
```
# Given values
```
a = 9
```

```
b = 2
```

```
# Calculate the result
result = operation(a, b)
```

```
# Expected result
expected_result = sp.Rational(3, 4) * sp.sqrt(3)
```

```
# Verify if the result matches the expected result
is_correct = sp.simplify(result) == sp.simplify(expected_result)
print(is_correct)
```
**Interpreter output**: **False**

