# OpenReview forum: "T1: Tool-integrated Verification for Test-time Compute Scaling in Small Language Models"
_ICLR.cc/2026/Conference — ICLR 2026 Poster_

### Official Review · Reviewer_tsxd · 2025-10-31

**Soundness:** 2
**Presentation:** 3
**Contribution:** 2
**Rating:** 4
**Confidence:** 4

**Summary:**

Tools can offload specific skills, such as arithmetic or fact retrieval, from language models to purpose-built systems. This paper extends that idea to verification: it introduces a tool-integrated verifier that allows smaller language models to act as effective verifiers. The method first applies a tool-based binary filter to discard incorrect candidates, and then uses a reward model to score the remaining answers. A simple theoretical model suggests that tool integration reduces the need for memorization, and experiments on GSM8K, MATH500 and MMLU-Pro show improved Best-of-N performance.

**Strengths:**

* Well written.
* Consistent empirical gains across several model families and sizes.
* Theoretical intuition connects tools to reduced memorization burden and to improved Best-of-N selection via reduced verifier noise.
* The fact that ToolV and GenRM stages are just LoRA adapters of the same model (which is the same as the reasoner) makes the deployment in low-resource scenarios easier.

**Weaknesses:**

1. I understand that offloading to tools allows smaller models, but the design choice seems odd: why restrict tool use to the verifier? Qualitative examples (e.g., Example G.1) show the verifier merely calls an equation solver that could equally have been used by the generator. Giving tools to the reasoner feels like the more direct and widely studied approach (e.g., Toolformert, PAL, TORA), so the novelty here mainly comes from an asymmetric setup rather than a new tool integration method.

2. It’s unclear if the improvements from Distilled GenRM → ToolV + Distilled GenRM arise from tool use or simply from adding another verification stage. A control experiment where the two-stage verifier is used but with no external tools would clarify whether the tools themselves drive the gains.
3. The proposed approach spends roughly 2× verification compute compared to the baseline, yet this is not accounted for. If Figures 3 and 4 plotted accuracy against the number of generated solutions, the ToolV + Distilled PRM curve should arguably be shifted right to reflect the extra verification cost. This additional cost might erase scaling gains in several cases. Without such normalization, it is difficult to judge compute-efficiency trade-offs fairly. Reporting wall-time, total output tokens, or at least the number of model calls would provide a more balanced comparison than using only the number of generations.

**Questions:**

Please provide the 2 suggested ablations above, and provide plots showing total cost in the x-axis, not only the number of generated solutions.

---

> ### Author Response · Authors · 2025-11-24
> **Response to Reviewer tsxd (1/2)**
>
> > W1. I understand that offloading to tools allows smaller models, but the design choice seems odd: why restrict tool use to the verifier? Qualitative examples (e.g., Example G.1) show the verifier merely calls an equation solver that could equally have been used by the generator. Giving tools to the reasoner feels like the more direct and widely studied approach (e.g., Toolformert, PAL, TORA), so the novelty here mainly comes from an asymmetric setup rather than a new tool integration method.
>
> **Tool use is placed in the verifier so that the framework remains compatible with discriminative reward models (e.g., PRMs) and avoids retraining the generator.** This asymmetric design is intentional and practically important.
>
> - Although tools could be used by the generator, doing so would require retraining or restructuring the generator to be tool-aware, which breaks compatibility with existing PRMs trained on standard generator formats. **Our two-stage design keeps the generator untouched so that PRMs remain directly usable without reformatting or retraining.**
> - Tool integration in verification is not new (e.g., Themis), but **our contribution is showing that two-stage verification setting enables a stronger small model in test-time scaling.** In particular, our results show that this design outperforms prior tool-integrated verifiers such as Themis.
> - The qualitative example (`Example G.1`) is a valid observation but also highlights the modularity: the generator need not call external tools because the verifier handles correctness checking with tools. This separation is precisely what allows small models to scale via verification without modifying or retraining the generator.
>
> In summary, our contribution does not come from simply adding tools but from demonstrating that this asymmetric design yields a practical and effective test-time scaling framework for small LMs that would not be possible with tool-augmented generation.
>
> ---
>
> > W2. It’s unclear if the improvements from Distilled GenRM → ToolV + Distilled GenRM arise from tool use or simply from adding another verification stage. A control experiment where the two-stage verifier is used but with no external tools would clarify whether the tools themselves drive the gains.
>
> We respectfully clarify that **the improvement does not come from simply adding another verification stage.**
> - A two stage GenRM without any tool integration does not improve accuracy. To isolate the effect of tool use, we compare two stage GenRM with ToolV plus GenRM under identical generator budgets on MATH500 with Llama-3.2-1B-Instruct. The results are shown below:
>
> | n  | One-stage GenRM | Two-stage GenRM | GenRM + ToolV |
> |----|-----------------|------------------|----------------|
> | 1  | 27.0            | 27.0             | 27.0           |
> | 2  | 31.6            | 29.6             | 33.8           |
> | 4  | 36.0            | 35.0             | 38.8           |
> | 8  | 40.0            | 39.8             | 43.8           |
> | 16 | 42.0            | 42.0             | 47.2           |
> | 32 | 44.8            | 44.4             | 49.4           |
> | 64 | 45.4            | 46.0             | 50.6           |
>
>
> - **The gains consistently come from ToolV.** Even when the amount of verification is doubled, a two stage GenRM without tools does not yield meaningful improvements. In contrast, the tool integrated verifier improves accuracy across all N. This confirms that the advantage is driven by tool use rather than by simply increasing the number of verification passes.
>
> We included this discussion and experimental results in `Table 5` and `Appendix E.7` of the revision.

---

> ### Author Response · Authors · 2025-11-24
> **Response to Reviewer tsxd (2/2)**
>
> > W3. The proposed approach spends roughly 2× verification compute compared to the baseline, yet this is not accounted for. If Figures 3 and 4 plotted accuracy against the number of generated solutions, the ToolV + Distilled PRM curve should arguably be shifted right to reflect the extra verification cost. This additional cost might erase scaling gains in several cases. Without such normalization, it is difficult to judge compute-efficiency trade-offs fairly. Reporting wall-time, total output tokens, or at least the number of model calls would provide a more balanced comparison than using only the number of generations.
>
> We thank the reviewer for raising this important point. We agree that verification cost should be considered, and we added a clarification and supporting statistics in the revision.
>
> - We report the total output tokens for GenRM verification (N=64) averaged by tasks:
>     - Generator outputs: 36,761
>     - GenRM verificaiton: 283,591
>     - ToolV code: 39,093
>     - Total: 359,446
>     - Verification compute increases with ToolV, but **the magnitude is small relative to the GenRM verification cost itself**, which is about 13.8% increases, not twice verification compute. This additional cost is limited and not large enough to change the qualitative scaling behavior.
> - For PRM verification, ToolV can be viewed as adding roughly 2 times verification compute as the reviewer mentioned. The comparison of 64 generations with PRM versus 32 generations with PRM and ToolV reflects this. **Even under this normalization, PRM and ToolV achieves higher accuracy** as in the below table.
>
> | Task    | Model        | PRM only (n=64) | ToolV + PRM (n=32) |
> | ------- | ------------ | ----------------- | -------------------- |
> | MATH500 | Llama-3.2-1B | 47.45             | 50.0                 |
> | MATH500 | Qwen2.5-0.5B | 46                | 46.6                 |
> | MATH500 | SmolLM2-360M | 5.6               | 8.0                  |
>
> - **Verification compute is not the issue, the use of compute is.** Prior work such as GenRM [1] already relies heavily on verification compute. Simply doubling verification passes without tools does not improve performance, as noted in W2. Our contribution is to show how to use verification compute effectively.
> - **Contirubtion is a practical and principled direction, not compute equalization.** We agree that our plots do not fully normalize compute. However, the goal of our framework is different. With a fixed generator budget, we show how additional verification compute can be allocated to a small model to produce more accurate final answers without relying on a larger model. We also provide theoretical support for this setting. This is the practical trade-off our method addresses, and we clarified this point in the revision.
>
> We again appreciate the reviewer for highlighting this. We included this discussion in `Section 6.4`, `Figure 10`, `Appendix E.6`, and `Figure 11` of the revision.
>
> [1] Zhang et al., Generative Verifiers: Reward Modeling as Next-Token Prediction, ICLR 2025

---

### Official Review · Reviewer_sFSt · 2025-11-01

**Soundness:** 3
**Presentation:** 2
**Contribution:** 3
**Rating:** 6
**Confidence:** 2

**Summary:**

The paper proposes Tool-integrated Verification (T1): a two-stage verifier for small LMs (sLMs) that (i) first filters candidate solutions using a tool-based verifier (ToolV) e.g., running generated Python or fact-checking with retrieved documents and (ii) then scores remaining candidates with a reward-model verifier (PRM or generative verifier). The core idea is to shift memorization-heavy verification to external tools so sLMs benefit more from test-time scaling (Best-of-N). The method is trained via multi-LoRA distillation from large teacher verifiers. Experiments on MATH500, GSM8K, and MMLU-Pro report that sLMs with T1 outperform or match much larger models, although some larger-model baselines are run with N = 1 while sLMs use N up to 64, which complicates fairness.

**Strengths:**

* Clear theoretical decomposition of sub-optimality into OTC and a verifier-driven factor; three-regime picture is intuitive and actionable. (Theorem 3.6; Fig. 1.)
* Precise AiC analysis with explicit coverage violation condition and sub-optimality formulas.
* SMC construction via maximal coupling with matching complexity to SRS; neatly links transport optimality with compute.

**Weaknesses:**

* Plots lack error bars/CI; seeds aren’t disclosed. Given the sampling-heavy setup, dispersion matters. Report mean ± sd over ≥3 seeds, plus 95% CIs or stratified bootstrap CIs for all main tables/figures.
* The binary-generator and simple addition check help intuition but are far from math word-problem or retrieval settings. The paper should tighten the stated scope and discuss where the guarantees plausibly transfer.
* We don’t see how accuracy moves with # ToolV code completions (k), # retrieved docs, verifier thresholds, or retrieval quality (BM25 vs dense, top-k). These directly test your causal story; please add these sweeps.

**Questions:**

* Cost/latency is missing for a method that leans on tools. Can you report tokens, wall-clock, and $ per instance for T1 vs PRM/GenRM-only under equal budgets?
* Please show accuracy vs k code completions (e.g., 1–8), vs # retrieved docs (1–5), and vs verifier thresholds. These ablations would isolate where T1 helps most.
* Which components of the theorems do you believe carry over to real math/RAG settings, and which do not? A brief “theory-to-practice” paragraph would calibrate reader expectations.

---

> ### Author Response · Authors · 2025-11-24
> **Response to Reviewer sFSt (1/2)**
>
> We appreciate the reviewer for the detailed and constructive feedback. The comments help sharpen both the empirical rigor and the theoretical communication of our work. Below we address each point and outline the revisions we made.
>
> ---
>
> > W1. Plots lack error bars/CI; seeds aren’t disclosed. Given the sampling-heavy setup, dispersion matters. Report mean ± sd over ≥3 seeds, plus 95% CIs or stratified bootstrap CIs for all main tables/figures.
>
> We agree that showing variance is important. Running multiple seeds is computationally expensive, and our current GPU resources limit the scale of these experiments. Still, we will report confidence intervals for several key figures during the rebuttal period to provide at least partial coverage.
>
> ---
>
> > W2. The binary-generator and simple addition check help intuition but are far from math word-problem or retrieval settings. The paper should tighten the stated scope and discuss where the guarantees plausibly transfer.
>
> > Q3. Which components of the theorems do you believe carry over to real math/RAG settings, and which do not? A brief “theory-to-practice” paragraph would calibrate reader expectations.
>
> **The examples in `Figure 1` and the simplified addition setting in theoretical analysis are intended as conceptual illustrations rather than direct claims of transfer to complex domains.** We will clarify this and tighten the stated scope.
>
> - The binary generator and simple addition testbed were included to build intuition for the role of tools within the two-stage design. They were not intended to imply one to one transfer to complex math or retrieval tasks.
> - To avoid over-claiming, we tightened the stated scope and explicitly note that the theoretical analysis are based on conceptual illustrations, not strict guarantees for all tool types or domains.
> - We added a theory-to-practice paragraph in `Appendix C.1` of the revision. This section clarifies the intention and scope of our theoretical analysis by explaining that the results are derived from simplified and idealized settings and are meant to illustrate conceptual points. The paragraph also clarifies that these analyses are not intended as direct models of practical verification behavior, but as conceptual motivations whose qualitative intuitions align with the empirical trends observed in our experiments.
>
> ---
>
> > W3. We don’t see how accuracy moves with # ToolV code completions (k), # retrieved docs, verifier thresholds, or retrieval quality (BM25 vs dense, top-k). These directly test your causal story; please add these sweeps.
>
> > Q2. Please show accuracy vs k code completions (e.g., 1–8), vs # retrieved docs (1–5), and vs verifier thresholds. These ablations would isolate where T1 helps most.
>
> We agree that these sweeps directly test our claims. Our current computational limits make it difficult to conduct the full set of experiments, but we will try our best to include several of these ablations within the rebuttal timeline.

---

> ### Author Response · Authors · 2025-11-24
> **Response to Reviewer sFSt (2/2)**
>
> > Q1. Cost/latency is missing for a method that leans on tools. Can you report tokens, wall-clock, and $ per instance for T1 vs PRM/GenRM-only under equal budgets?
>
> Thank you for raising this point. Wall clock latency depends on infrastructure, batching, and implementation details, so it is difficult to interpret fairly. Instead, we report token level cost, which directly corresponds to FLOPs and API cost. Under this comparison, **ToolV adds only a small overhead, and the tool integrated pipeline is more cost effective than GenRM only settings even under equal token budgets.**
>
> - Our pipeline consists of three components: solution generation (574.39 tokens), verification generation (4431.11 tokens), and code generation (610.84 tokens). **The total additional cost from using ToolV is small compared to verification generation**, which dominates the overall cost.
> - To compare ToolV + GenRM with GenRM-only under a matched budget, let us consider the number of verification generations $n$. GenRM-only requires 5005.5 tokens per sample, while ToolV + GenRM requires 5616.34 tokens. **This corresponds to a 1.12 multiplier.** In other words, GenRM-only would need to scale its number of generations by $k$ where $k \times 5005.5 = 5616.34$.
> - Thus, ToolV + GenRM with $n = 32$ is equivalent in cost to GenRM-only with $n \approx 36$. However, in practice, ToolV + GenRM with $n = 32$ already surpasses GenRM-only with $n = 64$ across most settings, as shown below:
>
> | Task    | Model        | GenRM only (n=64) | ToolV + GenRM (n=32) |
> | ------- | ------------ | ----------------- | -------------------- |
> | MATH500 | Llama-3.2-1B | 45.4             | **49.4**                 |
> | MATH500 | Qwen2.5-0.5B | 44.6                | **45.0**                 |
> | MATH500 | SmolLM2-360M | 7.0               | **8.4**                  |
> | GSM8K   | Llama-3.2-1B | 71.87             | **73.62**                |
> | GSM8K   | Qwen2.5-0.5B | 66.03             | **66.57**                |
> | GSM8K   | SmolLM2-360M | 31.39             | **36.85**            |
>
> - Therefore, under an equal-token budget, **ToolV + GenRM provides strictly better accuracy than GenRM-only.** Even with a smaller $n$, our tool-integration method achieves higher performance, indicating that the improvement is not from using more tokens but from leveraging external tools.
>
> We included this ablation in `Section 6.4` of the revision.

---

### Official Review · Reviewer_qnQz · 2025-11-01

**Soundness:** 3
**Presentation:** 3
**Contribution:** 2
**Rating:** 4
**Confidence:** 3

**Summary:**

This paper proposes Tool-integrated Verification (T1), a two-stage framework that uses external tools (e.g., code interpreters) to filter candidate solutions before sLM-based verification, offloading memorization-heavy tasks like numerical calculations. The authors provide theoretical analysis showing tools reduce memorization requirements and prove the two-stage design improves test-time scaling performance. Experiments demonstrate that T1 enables a Llama-3.2 1B model to outperform Llama-3.1 8B on MATH benchmarks, improving verification accuracy for both PRMs and generative verifiers.

**Strengths:**

The paper enables small language models to be reliable verifiers by combining it with the tools and boosts their performance by test-time scaling. The paper also provides solid theoretical proofs for the method, not only to well explain the method in the paper, but also to show a general analysis foundation for future research. The experiment design is also comprehensive, where the proposed T1 is compared with Majority Voting, Zero-shot GenRM, Distilled GenRM and Themis. The performance improvements are also remarkable on Math500 and GSM8K.

**Weaknesses:**

- Limited novelty and narrow focus: The core idea of using code interpreters for mathematical verification is well-established (PAL, Toolformer). The contribution is mainly combining existing techniques in the verification stage rather than introducing new methods. The paper focuses almost exclusively on mathematical reasoning (MATH, GSM8K), a narrow domain where code execution naturally provides ground truth verification. Even for MMLU-Pro tasks in the appendix, that retrieval helps verification is unsurprising and well-known from prior RAG literature.
- Missing key ablation: The paper fails to evaluate ToolV alone as a verifier, only ToolV + RM combinations. This makes it hard to assess the individual contribution of each stage or determine if the two-stage design is necessary versus using ToolV alone.
- Missing cost-benefit analysis: No computational overhead reported for generating and executing code or retrieval, unclear if performance gains justify additional inference cost versus using a larger model.

**Questions:**

1. What’s the exact performance of the proposed method and baselines in the chart? Could you provide some statistics at least for some key points?
2. The MATH500 and GSM8K are fairly easy benchmarks. Could you try the more challenging benchmarks such as AIME 24, AIME 25 (also math domain)? Perhaps the method will shine better for challenging tasks.
3. As mentioned in the weakness section, could you add the ablation and the cost-benefit analysis?
4. Theoretically, the proposed method is not limited to the small LMs. Will the performance improvement vanish if the method is based on larger LMs?

---

> ### Author Response · Authors · 2025-11-24
> **Response to Reviewer qnQz (1/2)**
>
> We appreciate the reviewer for the thoughtful and constructive feedback. We are glad that the reviewer found our theoretical proofs are solid and the experiment design is comprehensive. We provide clarifications for most concerns arise in below.
>
> > W1. Limited novelty and narrow focus: The core idea of using code interpreters for mathematical verification is well-established (PAL, Toolformer). The contribution is mainly combining existing techniques in the verification stage rather than introducing new methods. The paper focuses almost exclusively on mathematical reasoning (MATH, GSM8K), a narrow domain where code execution naturally provides ground truth verification. Even for MMLU-Pro tasks in the appendix, that retrieval helps verification is unsurprising and well-known from prior RAG literature.
>
> **We would like to clarify that our contribution is not about using tools for verification.**
> The core idea is the design of the verifier system that enables effective test-time scaling with small models. This is a different goal from prior works such as PAL and Toolformer, which integrate tools directly into the generation process and rely on large models.
>
> - **Our two-stage verifier design fundamentally changes how verification is performed with small models.** By separting generation and verification in test-time scaling, tool use becomes an isolated component of a structured verification pipleline. This design allows small verifiers to scale their accuracy with the number of samples, which has not been addressed in prior tool integration works.
> - The two stage structure enables practical compatibility with existing verifier models such as PRMs. Since the verifier consumes solutions in a fixed format, PRMs trained for prior generators can be reused by simple distillation. This property is important in test-time scaling pipelines and is not supported by tool-integrated generator methods. **We emphasize that this compatibility is a direct consequence of the separation between generation and verification.**
> - Our evaluation domains follow the standard practice in recent test-time scaling works [1,2]. GSM8K and MATH500 are the benchmarks used in prior works since they provide clear and repoducible measurement for scaling behavior with small models. **Our goal is to answer how to build a stronger small model verifier under this setting.**
>
> In summary, the focus of this work is not the use of tools but the verifier framework that makes test-time scaling effective with small models, together with its practical implications for PRM reuse and reproducibility.
>
> [1] Song et al., Mind the gap: Examining the self-improvement capabilities of large language models, ICLR 2025
>
> [2] Liu et al., Can 1b llm surpass 405b llm? rethinking compute-optimal test-time scaling, Preprint
>
> ---
>
> > W2. Missing key ablation: The paper fails to evaluate ToolV alone as a verifier, only ToolV + RM combinations. This makes it hard to assess the individual contribution of each stage or determine if the two-stage design is necessary versus using ToolV alone.
>
> **ToolV alone is a stong verifier, but it does not capture the full range of failure modes that arise in small model generations.** ToolV mainly validates executable steps such as calculations, while PRMs and GenRMs provide copmlementary preferences that detect reasoning-level erros that are not executable. This is why the two-stage design is necessary.
>
> - We provide the missing ablation below with Llama-3.2-1B-Instruct model. ToolV alone outperforms PRM and GenRM, but the combined two-stage design consistently achieves the best performance.
>
> | MATH500, PRM       | Accuracy |
> |---|---|
> | PRM      | 47.4     |
> | ToolV    | 52.4     |
> | ToolV + PRM  | 53.2     |
>
> | GSM8K, GenRM       | Accuracy |
> |---|---|
> | GenRM     | 71.87     |
> | ToolV     | 73.62     |
> | ToolV + GenRM  | 74.60     |
>
> - These results show that ToolV improves verification, but it does not subsume the role of reward models. **The two-stage structure is necessary because each stage covers different types of verification signals.** We added this ablation and discussion to `Appendix E.5` of the revision.

---

> ### Author Response · Authors · 2025-11-24
> **Response to Reviewer qnQz (2/2)**
>
> > W3. Missing cost-benefit analysis: No computational overhead reported for generating and executing code or retrieval, unclear if performance gains justify additional inference cost versus using a larger model.
>
> Thank you for raising this point. We acknowledge that our initial submission did not fully discuss the computational overhead of ToolV. We would like to clarify that (1) **enabling small models to serve as effective verifiers provides unique benefits**, and (2) even when accounting for the additional computation, **ToolV remains clearly advantageous**.
>
> - Using a larger verifier directly increases GPU memory requirements. For this reason, comparing a large verifier with a small verifier is not directly meaningful. Making small models strong verifiers offers clear benefits for memory-constrained scenarios such as on-device or limited GPU settings.
> - Even when considering the computational cost of ToolV, the overhead is minimal. We analyzed the average number of generated tokens per solution for each stage following the recent work [1]:
>     1. solution generation: 574.39 tokens
>     2. verification generation: 4431.11 tokens
>     3. code generation: 610.84 tokens
> - In total, this results in 5616.34 tokens per solution. If we compare this to using a $n$ times larger verifier (ignoring the additional prefill cost that larger models would incur),
>     - $5616.34$ vs. $574.39 + 4431.11 * n$ gives $n = (5616.34 - 574.39) / 4431.11 = 1.14$.
>     - In other words, the computational **cost of ToolV is roughly equivalent to using a $1.14$ times larger verifier**, which is a small overhead.
> - Considering that ToolV 1B + GenRM 1B outperforms GenRM 8B on the MATH benchmark (`Figure 7`), the performance gain overwhelmingly justifies this small additional inference cost compared to simply scaling up the verifier.
>
> We have included this discussion in `Section 6.4` of the revision.
>
> [1] Singhi et al., When To Solve, When To Verify: Compute-Optimal Problem Solving and Generative Verification for LLM Reasoning, COLM 2025
>
> ---
>
> > Q1. What’s the exact performance of the proposed method and baselines in the chart? Could you provide some statistics at least for some key points?
>
> Thank you for raising this point. We added `Tables 8 to 16` in the Appendix, which correspond to `Figures 3, 4, and 5`, and included an exact mapping and detailed descriptions in `Appendix E.8` of the revision.
>
> > Q2. The MATH500 and GSM8K are fairly easy benchmarks. Could you try the more challenging benchmarks such as AIME 24, AIME 25 (also math domain)? Perhaps the method will shine better for challenging tasks.
>
> AIME is difficult for small models we used because the generator accuracy is significantly low. Our focus is on test time scaling scenarios where high quality candidate solutions are available, so that the verifier can meaningfully differentiate correct and incorrect outputs. In contrast, AIME provides very few correct candidates for the small models we study. For example, even with N=64 majority voting, the strongest generator in our setting (Llama-3.2-1B-Instruct) reaches only 6.7% accuracy.
>
> > Q3. As mentioned in the weakness section, could you add the ablation and the cost-benefit analysis?
>
> We added required ablations and cost-benefit analysis in `Appendix E.5` (ablation) and `Section 6.4` (cost-benefit analysis) of the revision. Thank you again for your great suggestion to improve the quality and clarity of our work.
>
> > Q4. Theoretically, the proposed method is not limited to the small LMs. Will the performance improvement vanish if the method is based on larger LMs?
>
> The method also improves larger verifiers, although the gains are smaller. The main goal is to enable small verifiers to reach the performance of large verifiers, and we achieve that in our experiments.
>
> - Larger verifiers benefit from ToolV, as shown in `Figure 7`, but the improvement is smaller because they already perform well.
> - Our objective is not to improve already strong large models, but to allow small verifiers to match them.
> - On MATH500, Llama 1B GenRM plus ToolV already surpasses Llama 8B GenRM, showing that the performance benefit does not vanish but is strongest in the small LM regime.

---

### Author Response · Authors · 2025-11-24
**General Response to Reviewers**

We thank the reviewers for their time, effort, and constructive feedback, which have significantly strengthened this work. We are encouraged that reviewers found our **theoretical proofs are solid and the experiment design is comprehensive** [qnQz], and noted that our simplified theoretical illustrations **help build intuition** for the proposed mechanism [sFSt]. We appreciate that the reviewers engaged deeply with our work, providing detailed suggestions regarding ablations and cost analysis that helped **sharpen both the empirical rigor and the theoretical communication** of our framework [sFSt, tsxd].

Below we summarize the key clarifications added during the rebuttal:

- **The two-stage verifier framework is a deliberate design for test-time scaling** [qnQz, tsxd]. We clarified that our contribution is not merely using tools, but designing a verification pipeline that enables small models to scale effectively. This framework design with the generator untouched ensures practical compatibility with existing PRMs without retraining of the generator.
- **ToolV and Reward Models are complementary and necessary** [qnQz, tsxd]. We presented new ablations showing that while ToolV validates executable steps, it cannot replace Reward Models which detect reasoning errors. The combined two-stage design outperforms either component alone, confirming that improvements stem from tool integration rather than simply adding verification stages.
- **ToolV remains effective under equal-compute budgets** [qnQz, sFSt, tsxd]. We provided a detailed cost analysis demonstrating that ToolV adds only a small token overhead with GenRM ($\approx 1.12\times$). Even when normalized for compute, our method consistently outperforms baselines that simply scale up the number of generations.

We also added clarifications addressing the specific questions raised by each reviewer. In addition, we updated the manuscript as follows:

- We added **`Section 6.4`** to provide a detailed **cost-benefit analysis**, comparing token-level overhead and demonstrating the efficiency of ToolV under equal-budget settings [qnQz W3, sFSt Q1, tsxd W3].
- We added **`Appendix E.5 & E.7`** to include **ablation studies** assessing ToolV alone vs. combined models [qnQz W2] and verifying that gains are driven by tool use [tsxd W2].
- We added **`Appendix E.8`** (Tables 8–16) to provide **exact performance statistics** corresponding to Figures 3, 4, and 5 [qnQz Q1].
- We added **`Appendix C.1`** to clarify the **scope of our theoretical analysis** and its "theory-to-practice" connection, ensuring the interpretation of our simplified testbeds is properly contextualized [sFSt W2, Q3].

We thank the reviewers again for the constructive insights that helped sharpen the scope, clarity, and positioning of this work.

---

### Meta-Review · Area_Chair_RxBp · 2025-12-05

**Summary:**

This paper proposes a two-stage framework to do test time scaling for small language models. The first stage uses a tool to filter candidates, followed by the verification stage using a reward model. The authors provide some theoretical analysis demonstrate that tool use reduces memorization requirements. Experiments demonstrate that this design can improve over with several baselines where tool was not used as a filter.

Reviewer concerns on missing baseline and ablations are mostly addressed from supplemental experiments, which suggest consistent results.

The paper claims the framework to be test time scaling for sLM for verification, however, it seems the overall setup is under the sLM, i.e. models that generate the solution is also an sLM. The authors should adjust their claims accordingly to test time scaling of small LMs with verification.

Writing clarity could be better, for example, there is a lot of use of the word "GenRM", where some times it meant none-distilled reward model, sometime it was from the distilled model. Clarify those occurrences would make the experiments much easier to interpret.

**Reviewer Concerns:**

All reviewers concerns on missing experiments were performed, with consistent results.
Experiments on more challenging math dataset, e.g. AIME 24/25 were not performed, where the authors mentioned that the generator is week. Hence the comment regarding the paper's claim should be on small LMs only, rather than method for scaling sLM verifier.

**Reviewer Scores:**

qnQz: 4 -> 5, points on novelty is hard to overturn
sFSt: 6 -> 6, reviewer confidence is low
tsxd: 4 -> 6, concerns mostly addressed

---

### Decision · Program_Chairs · 2026-01-26

Accept (Poster)